# Two-stage optimization based on heterogeneous branch fusion for knowledge distillation

**Gang Li**[1]☉, **Pengfei Lv**[1]☉, **Yang Zhang**[2]*, **Chuanyun Xu**[2]*, **Zihan Ruan**[1], **Zheng Zhou**[1], **Xinyu Fan**[1], **Ru Wang**[1], **Pan He**[2]

**1** School of Artificial Intelligence, Chongqing University of Technology, Chongqing, China, **2** School of Computer and Information Science, Chongqing Normal University, Chongqing, China

☉ These authors contributed equally to this work.

* zhangyang@cqnu.edu.cn

**Data availability statement:** The data underlying his study have been uploaded to GitHub and are accessible using the following link: https://github.com/GitLpf/THFKD.

## Abstract

Knowledge distillation transfers knowledge from the teacher model to the student model, effectively improving the performance of the student model. However, relying solely on the fixed knowledge of the teacher model for guidance lacks the supplementation and expansion of knowledge, which limits the generalization ability of the student model. Therefore, this paper proposes two-stage optimization based on heterogeneous branch fusion for knowledge distillation (THFKD), which provides appropriate knowledge to the student model in different stages through a two-stage optimization strategy. Specifically, the pre-trained teacher offers stable and comprehensive static knowledge, preventing the student from deviating from the target early in the training process. Meanwhile, the student model acquires rich feature representations through heterogeneous branches and a progressive feature fusion module, generating dynamically updated collaborative learning objectives, thus effectively enhancing the diversity of dynamic knowledge. Finally, in the first stage, the ramp-up weight gradually increases the loss weight within the period, while in the second stage, consistent loss weights are applied. The two-stage optimization strategy fully exploits the advantages of each type of knowledge, thereby improving the generalization ability of the student model. Although no tests of statistical significance were carried out, our experimental results on standard datasets (CIFAR-100, Tiny-ImageNet) and long-tail datasets (CIFAR100-LT) suggest that THFKD may slightly improve the student model's classification accuracy and generalization ability. For instance, using ResNet110-ResNet32 on the CIFAR-100 dataset, the accuracy reaches 75.41%, a 1.52% improvement over the state-of-the-art (SOTA).

## Introduction

Deep neural networks (DNNs) have demonstrated powerful performance in numerous computer vision tasks, such as image classification [1–4], object detection [5–7], and semantic

**Funding:** This study was funded by the China Chongqing Municipal Science and Technology Bureau, grant number CSTB2024TIAD-CYKJCXX0009 (Y. Z.), CSTB2024NSCQ-LZX0043 (C. X.); Chongqing Municipal Commission of Housing and Urban-Rural Development,grant number CKZ2024-87 (Y. Z.); the Chongqing University of Technology graduate education high-quality development project, grant number gzlsz202401 (G. L.); the Chongqing University of Technology - Chongqing LINGLUE Technology Co.,Ltd.. Electronic Information (artificial intelligence) graduate joint training base (G. L.); the Postgraduate Education and Teaching Reform Research Project in Chongqing, grant number yjg213116 (G. L.); and the Chongqing University of Technology - CISDI Chongqing Information Technology Co., LTD. Computer Technology graduate joint training base (C. X.). This study does not involve any ethical issue.

segmentation [8,9]. However, high-performance models often come with substantial parameter counts and computational costs, making them difficult to deploy in scenarios requiring lightweight models. Researchers have developed various model compression and acceleration techniques [10]. As a representative approach in model compression and acceleration, knowledge distillation (KD) [11] effectively balances model size and performance by transferring knowledge from a large teacher model to a compact student model.

Existing knowledge distillation methods are generally classified into offline knowledge distillation (KD) and online knowledge distillation (OKD) [12]. As shown in Fig 1(b), in the KD method, a two-stage training approach is used: first, a large teacher model is pre-trained, and then the distilled knowledge is transferred to a smaller student model, encouraging the student to learn the complex knowledge of the teacher. However, in practice, it is difficult for the student to thoroughly learn the knowledge provided by the teacher due to the significant capability gap between the converged teacher model and the student model trained from scratch [13]. Furthermore, the fixed knowledge of the teacher model cannot fully enhance the student's generalization ability. The dynamic information from the training process can be utilized as a source of knowledge [14], and increasing the diversity of dynamic information can effectively supplement and expand knowledge.

The OKD method uses a single-stage training approach to directly optimize the student model, continuously updating knowledge during training and fully allowing the student to utilize the rich information from multiple outputs. For example, Deep Mutual Learning (DML) [15] extracts knowledge by training multiple student models with the same structure in parallel and having them learn from each other. Alternatively, as shown in Fig 1(a), the Multi-branch OKD approach, proposed by On-the-Fly Native Ensemble (ONE) [16], employs a single model with multiple parallel branches for training, utilizing the average prediction generated by the branches as the learning objective, achieving better performance than DML. The OKD method narrows the capability gap between models, making knowledge transfer easier. However, it lacks reliable supervision information early in the training process, leading to the optimization direction being dominated by incorrect information [17]. At the same time, a pre-trained teacher can provide stable and comprehensive guidance.

Although both KD and OKD have shortcomings, Liu et al. [18] argue that they complement the knowledge distillation problem. By evaluating Similarity of neural network representations revisited (CKA) [19], imitation error rate (IER), and misguidance rate (MR), they demonstrated that the training strategy of OKD is more effective than the static knowledge transfer in KD. However, the pre-trained teacher model in KD can provide more reliable supervision information to the student. In early learning, the teacher's cognition significantly influences the student group, which later exchanges and shares their different insights with other students, further expanding knowledge. However, a simple combination cannot fully leverage the comprehensiveness of static knowledge, and dynamic knowledge also lacks diversity.

Based on the above findings, this paper proposes a Two-stage optimization based on heterogeneous branch fusion for knowledge distillation (THFKD), which fully exploits the advantages of both types of knowledge through a two-stage optimization strategy. First, the teacher's knowledge guides the student in global learning while the student generates collaborative learning objectives based on heterogeneous branches and a progressive feature fusion module, thus transferring diversified, dynamic knowledge. Then, the ramp-up weight is used to control weight loss at different stages, implementing the two-stage optimization. In the early stage of training, the static knowledge of the teacher guides the complete learning of the student, while the diversified, dynamic knowledge becomes more important in the later stages. In the testing phase, only the target branch is retained, with the remaining branches

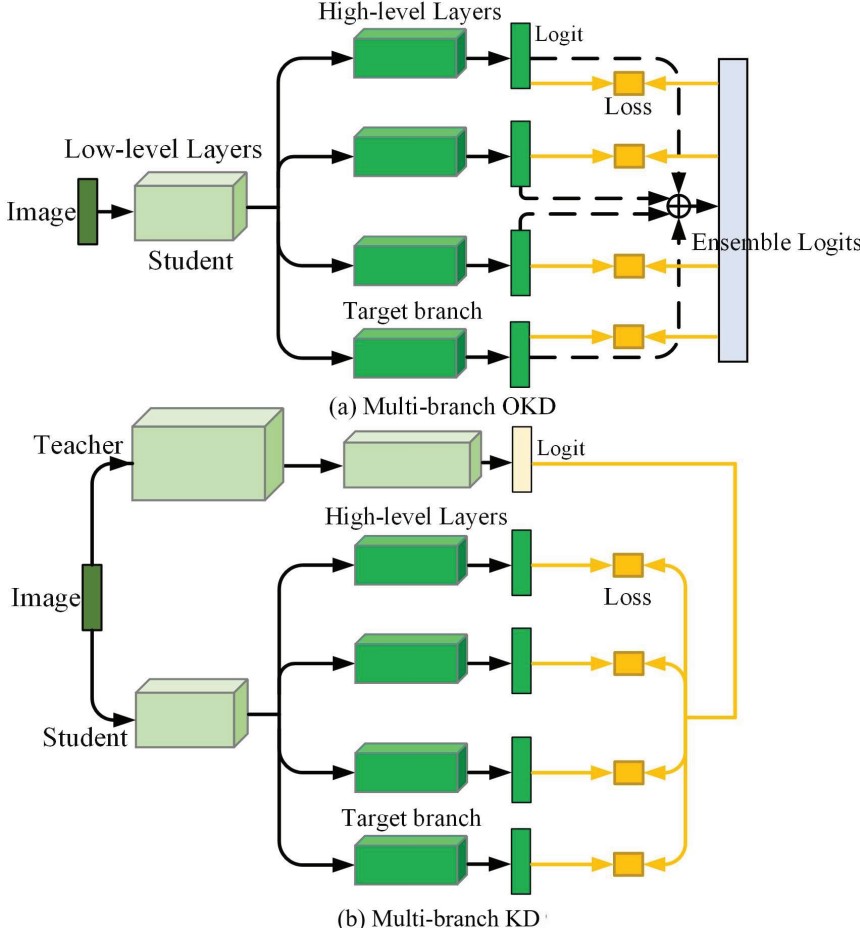

**Fig 1. Two training frameworks.** (a) using the OKD method and (b) using the KD method.

serving as training aids without incurring additional costs, or other branches can be selectively retained according to actual needs. Experimental results show that the integration of multiple branches can improve performance.

The main contributions of this paper are summarized as follows:

1. We propose a novel knowledge distillation training framework. Unlike conventional approaches, our framework enables the student model to learn not only from a pre-trained teacher model but also to acquire dynamic knowledge from heterogeneous branches. The training process consists of two stages: first establishing foundational cognition through comprehensive static knowledge, then introducing diversified dynamic knowledge to achieve synergistic optimization between static and dynamic knowledge.

2. We introduce heterogeneous branches and a Progressive Feature Fusion (PFF) module, which endows the branches with distinct abilities while efficiently fusion differentiated feature representations, thereby enhancing the diversity of dynamic knowledge.

3. Extensive comparative experiments demonstrate the superior performance of the proposed THFKD across various datasets and network architectures. For instance, on Tiny-ImageNet, THFKD achieves a 0.71% improvement over state-of-the-art (SOTA) methods.

## Related work

### Methods based on static knowledge

Initially proposed by Hinton et al. in 2015 [11], Knowledge Distillation (KD) is generally considered an efficient model compression technique. Its core idea is to use a larger teacher model to guide the training of a smaller student model. In subsequent research, scholars have proposed various forms of teacher knowledge, enabling students to mimic different aspects of the teacher, mainly divided into softened logits and intermediate features. For example, earlier KD methods encourage students to mimic the logits of the teacher softened by temperature [9,11,15,20,21]. However, while Knowledge Distillation (KD) ensures the stability of knowledge transfer by leveraging supervision from a pre-trained teacher model, the static nature of such knowledge inherently constrains the generalization capability of the student model. The current state-of-the-art methods are based on intermediate feature knowledge [22–29]. For instance, FitNet [28] directly transfers the teacher's features to the student, while other approaches transfer the correlations between samples captured by the teacher to the student [23]. Additionally, NRKD (A new similarity-based relational) [30] transfers knowledge of neighborhood relationships by selecting the K-nearest neighbors for each sample based on similarity metrics. ReviewKD (via knowledge review) [29], first proposed matching feature information through cross-stage connection paths. However, most intermediate feature-based knowledge distillation methods require consistent feature resolutions between the teacher and the student. This requirement implies that the teacher and student must either share the same model structure or rely on complex feature alignment modules [31]. Therefore, this study utilizes the softened logits to jointly learn static and dynamic knowledge rather than solely relying on the teacher model.

### Methods based on dynamic knowledge

Deep Mutual Learning (DML) [15] proposed an online knowledge distillation method that does not require a pre-trained teacher model, using the simultaneous training of multiple student models to enable them to learn from each other. To reduce training costs and complexity, ONE [16] uses a single multi-branch model and employs a gating module to aggregate branch predictions, generating a dynamic teacher for knowledge transfer. However, multi-branch architectures suffer from homogenization issues due to insufficient diversity in the acquired knowledge across branches. Subsequent research has further enhanced the stability and diversity of online distillation. For example, OKDDip (Online knowledge distillation with diverse peers) [32] utilizes a self-attention mechanism to derive a separate target for auxiliary peers, increasing diversity during training and employing second-level distillation to transfer the knowledge from aggregated predictions to the target student. PCL (Peer collaborative learning) [33] uses the temporal average model of peers as an ensemble teacher and transfers knowledge among students through two different ensemble strategies, making the optimization more stable. However, the student model is susceptible to noise in the early stages of training, which may lead to the aggregation of incorrect information in the dynamic knowledge. In the later stages, dynamic knowledge can further optimize the model, so the loss weights are adjusted to control the impact of dynamic knowledge at different stages.

### Multi-branch architectures and feature fusion

Compared to training multiple independent models in DML [15], ONE [16] uses a multi-branch model architecture. This architecture reduces the parameters and avoids the asynchronous updates of independent models, simplifying the distillation process. However, in

multi-branch architectures, the knowledge acquired by each branch during the training process suffers from limited diversity, resulting in a homogenization problem that ultimately constrains the model's performance. This phenomenon occurs because parallel branches tend to converge toward similar feature representations under shared optimization objectives. Additionally, feature fusion can effectively utilize the feature information from each branch, further enhancing the representational capability of the model. For example, FFL (Feature fusion for online mutual knowledge distillation) [34] applies multiple pointwise convolutions to features fused along the channel dimension, which applies to any model architecture and improves the performance of both the fused classifier and each student model. AFF (Attentional feature fusion) [35] focuses on feature information from different dimensions and assigns different weights to this information using an attention mechanism. However, most methods remain constrained by the absence of efficient fusion mechanisms, leading to suboptimal utilization of inter-branch information. Compared to traditional feature fusion methods, Concat results in many parameters due to linear combinations, while average fusion ignores differences in feature contributions. Our proposed heterogeneous branches differ from the target branch regarding model depth and feature representation. The progressive feature fusion module, designed with a spatial attention mechanism and a hierarchical progressive structure, decomposes the global fusion problem into multiple local fusion problems, thereby effectively improving feature utilization efficiency and feature quality.

## Preliminary

### Notations

The student model adopts a multi-branch architecture where each branch shares parameters in the low-level layers but has independent parameters in the high-level layers, functioning as multiple parallel-trained students. Branch-1 is designated as the target branch, forming the baseline model together with the low-level layers. We denote the total number of branches as $B$, the branch index as $b \in (1, 2, ..., B)$, and the features generated by the branch as $F_b$.

Given a dataset with $K$ classes, where $k \in (1, ..., K)$, and a training sample $x$ with its corresponding one-hot label $y = [y_1, ..., y_K]$, where $y_1, ..., y_K \in \{0, 1\}$ and $\sum_{k=1}^{K} y_k = 1$, when $x$ is input into the student model, the output probability of the $b$-th branch for class $k$ is calculated as:

$$p_k^b = \frac{\exp\left(z_k^b\right)}{\sum_{i=1}^{K} \exp\left(z_i^b\right)} \tag{1}$$

where $z^b = [z_1^b, ..., z_K^b]$ is the logit of the $b$-th branch. For multi-class classification tasks, training is performed by minimizing the cross-entropy loss between the predicted vector and the ground-truth labels:

$$L_{ce}^b = -\sum_{i=1}^{N} \sum_{k=1}^{K} y_{i,k} \log p_{i,k}^b \tag{2}$$

where $N$ represents the number of samples, $y_{i,k}$ is the ground-truth label (one-hot encoded) of sample $i$, and $p_{i,k}$ is the predicted probability by the model that sample $i$ belongs to class $k$.

### Knowledge distillation

The core of traditional knowledge distillation (KD) methods is to use the softened teacher outputs as the supervisory signal for training the student model. The formulas for the softened

teacher and student output probabilities are as follows:

$$\widetilde{p}_k^j = \frac{\exp\left(z_k^j/\tau\right)}{\sum_{i=1}^K \exp\left(z_i^j/\tau\right)}, j \in \{t, s\} \tag{3}$$

where $\tilde{p}_k^t$ and $\tilde{p}_k^s$ represent the softened outputs of the teacher and student, with $\tau$ being the temperature parameter. Then, the Kullback-Leibler divergence between $\tilde{p}_k^t$ and $\tilde{p}_k^s$ is minimized, and the distillation loss is calculated as follows:

$$L_{kl} = KL\left(\widetilde{p}^t, \widetilde{p}^s\right) = \sum_{k=1}^K \widetilde{p}_k^t \log \frac{\widetilde{p}_k^t}{\widetilde{p}_k^s} \tag{4}$$

the total loss for knowledge distillation is calculated as follows:

$$L_{kd} = L_{ce} + \tau^2 L_{kl} \tag{5}$$

As the temperature increases, the gradient of the distillation loss decreases. Therefore, $L_{kl}$ is multiplied by $\tau^2$ to ensure that the gradient contributions of the cross-entropy loss and the distillation loss remain roughly consistent despite changes in temperature [11].

## Proposed method

In this section, we introduce the overall framework of Two-stage optimization based on heterogeneous branch fusion for knowledge distillation (THFKD), including the detailed implementation of the heterogeneous branches and the Progressive Feature Fusion module, as well as the loss function of THFKD.

### Heterogeneous branch structure

The student model is trained using a multi-branch structure. However, due to the consistent construction among branches, each student's capabilities are identical, leading to the homogenization problem during training, which hinders the diversification of dynamic knowledge. Therefore, this study designs heterogeneous branches by adding additional auxiliary blocks to the target branch structure to construct students with different capabilities. As shown in the Heterogeneous Branch section of Fig 2, Branch-1 is set as the target branch. In contrast, other branches gradually add auxiliary blocks with $C_A = A \cdot C(A = (1, 2, \dots, b - 1))$ channels, where $C$ is the number of channels in the high-level layers. Thus, the channel count of the auxiliary blocks increases in linear multiples, aiming to capture differentiated feature maps and providing a foundation for the diversity of collaborative learning. Let $F_b^C$ denote the feature map input to the auxiliary block, where $C$ represents the number of channels in the feature map. The specific formulation is as follows:

$$F_b^{C_A} = \text{Conv}1\left(F_b^C\right), F_b^{C_A} = \text{Conv}2\left(F_b^{C_A}\right), F_b = \text{Conv}2\left(F_b^{C_A}\right) + F_b^C \tag{6}$$

Specifically, the auxiliary blocks are encapsulated as a basic unit that primarily includes two $1 \times 1$ convolutional layers and one $3 \times 3$ convolutional layer. Where $Conv1$ is employed to increase the number of channels $(C - C_A)$, $Conv2$ extracts higher-level feature information through the expanded channels, and $Conv3$ uniformly restores the channel count to $C$. The

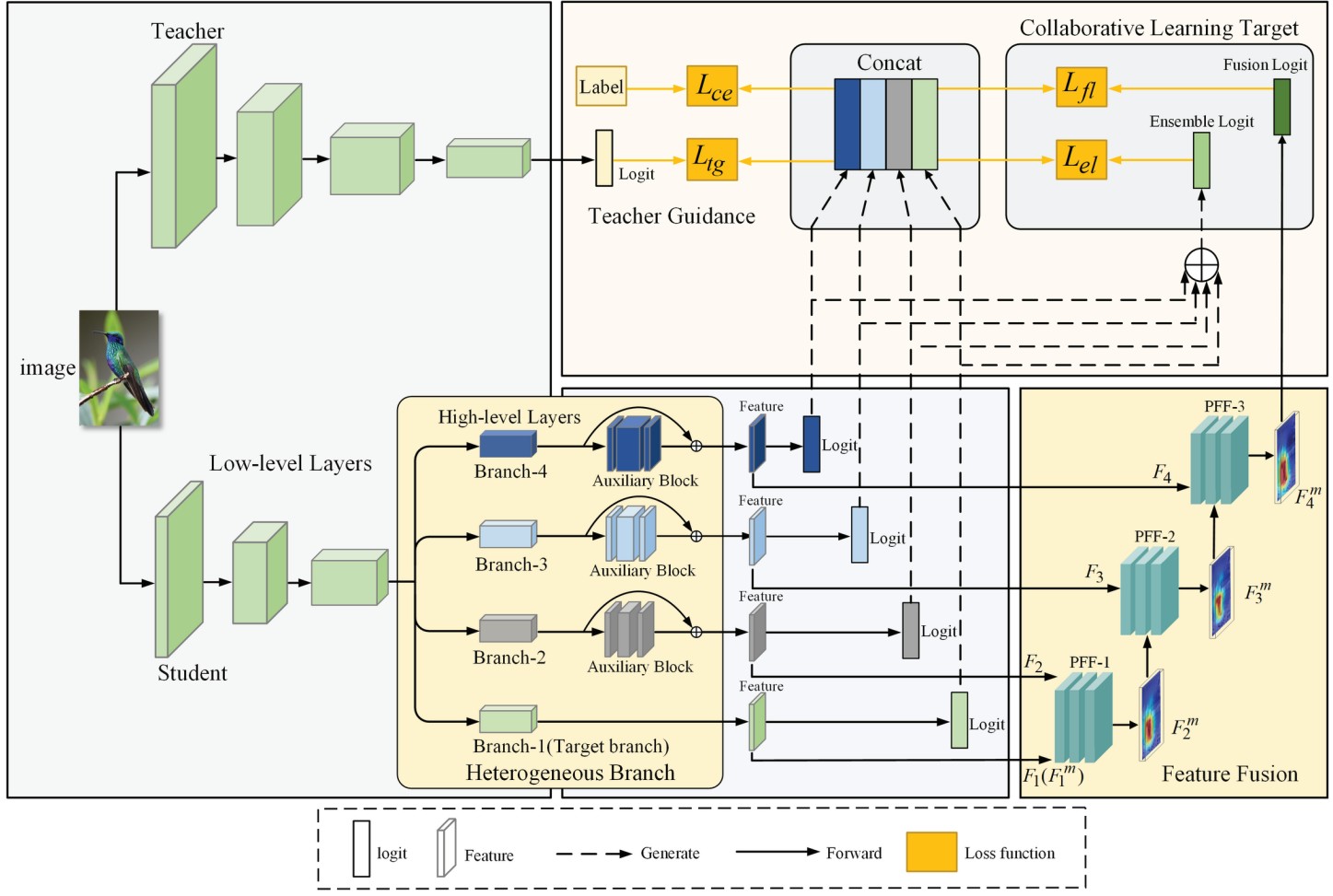

**Fig 2. An overview of two-stage optimization based on heterogeneous branch fusion for knowledge distillation (THFKD).** (1) It consists of pre-trained teacher and multi-branch student models. The teacher model guides each branch to form the $L_{tg}$ loss. (2) The student model constructs heterogeneous branches by adding auxiliary blocks and introduces PFF for progressive fusion. The Collaborative Learning Target comprises Fusion Logit and Ensemble Logit, with losses denoted as $L_{fl}$ and $L_{el}$, respectively. (3) The logits of the four branches are concatenated along the branch dimension, and the loss for each branch is computed concerning the teacher logit, collaborative learning target, and labels.

output is then combined with the original feature map $F_b^C$ via a residual connection to mitigate overfitting. Auxiliary blocks with a gradually increasing number of channels are added to the branches, creating differences in receptive field expansion and feature dimensions. This design forces different branches to learn complementary feature representations: shallow branches learn the fundamental features of samples, while deep branches capture subtle differences.

For a more straightforward presentation, we take ResNet32 architecture, for example, to elaborate on how to add the auxiliary blocks. The ResNet model is structured into three stages: the first two stages are low-level layers shared by each branch, and after the high-level layers in the third stage, auxiliary blocks are added to construct heterogeneous branches. In ResNet32, the number of channels in the third stage is 64, so the auxiliary blocks added to the $b$-th branch have $(b-1) \times 64$ channels.

## Progressive feature fusion module

Efficient fusion methods can enhance the utilization of features. AFF [35] suggests that the initial quality of feature fusion profoundly impacts the final fusion weights when attention mechanisms are introduced. Therefore, feature fusion modules can recursively integrate received features with another fusion module [8]. The heterogeneous branches designed in this work increase the network depth and capture feature information with different receptive fields by multiplying the number of channels, providing a rich feature basis for the fusion of global and local information. As shown in Fig 2, the Feature Fusion section includes three modules: PFF-1, PFF-2, and PFF-3. THFKD employs a progressive fusion approach to gradually integrate features, effectively improving fusion quality compared to simultaneously fusing features from $B$ branches.

The Progressive Feature Fusion (PFF) module is shown in Fig 3. First, the features $F_b$ from the $b$-th branch are initially fused with the fused features $F_{b-1}^m$ from the previous level. The initial fused features are then processed through two paths: one path includes two $1 \times 1$ convolutional layers to capture local information. In contrast, the other path employs an additional global pooling layer to capture global information. A Sigmoid function activates the features from both paths to obtain weight scores. These weight scores are then used to multiply the features $F_b$ and $F_{b-1}^m$, with parameter $g$ adjusting the importance of each branch, $g$ is a scalar parameter with a value range of 0 to 1, when g approaches 1, the current branch features dominates the fusion output, when g approaches 0, the fused features from the previous level receive higher weight. In our method, g is set to 0.5, ensuring equal importance of both current branch features and prior fused features, ultimately resulting in the fused features for this level.

The feature map of the first branch is used as the initial input fused feature, i.e., $F_1^m = F_1$. The expression for the features outputted by subsequent progressive fusion is:

$$F_b^m = D\left(F_b, F_{b-1}^m\right), b \in \{2, \dots, B\} \tag{7}$$

$$D\left(F_b, F_{b-1}^m\right) = g\left(F_b + F_{b-1}^m\right) \times F_b + (1 - g)\left(F_b + F_{b-1}^m\right) \times F_{b-1}^m \tag{8}$$

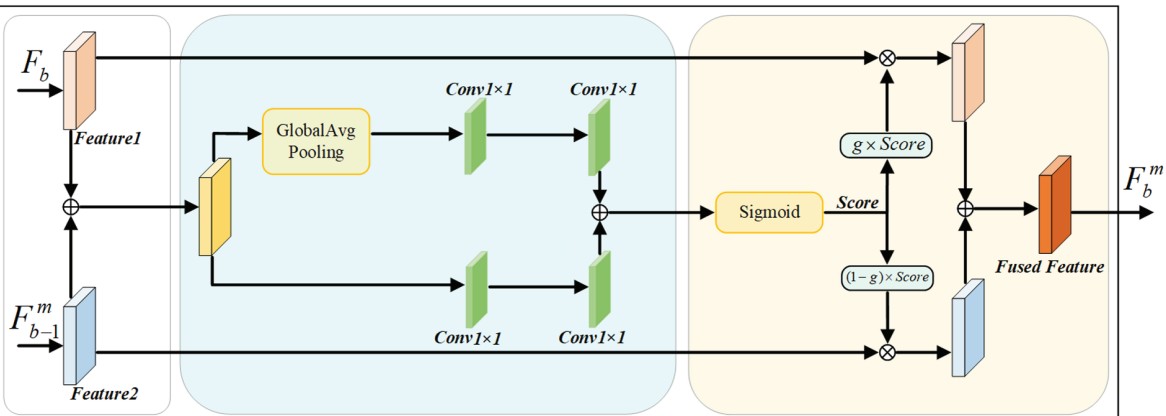

**Fig 3. Progressive feature fusion module.** The incoming features are processed through two distinct paths, and an attention mechanism (sigmoid) generates weight scores (score) to capture important information.

where $D(\cdot)$ is the fusion function from the Progressive feature fusion module, which is iteratively utilized throughout the training process to progressively amalgamate features. For instance, with the number of branches in the model set to four, the ultimate fused feature is denoted as $F_4^m$, where the fusion process is articulated as $F_4^m = D(F_4, D(F_3, D(F_2, F_1^m)))$.

## Collaborative learning target

This study designs a collaborative learning target to transfer diversified dynamic knowledge effectively. The collaborative learning targets consist of two parts: (1) Fusion Logits, which are obtained from the fused features $F_B^m$; and (2) Ensemble Logits, which are derived from the integration of branch logits. The Fusion Logits and Ensemble Logits are then softened using Eq (3) to obtain $\widetilde{p}_c^{m(B)}$ and $\widetilde{p}_c^e$, respectively. The two types of distillation losses are calculated as follows:

$$L_{el} = \tau^2 \sum_{b=1}^{B} KL\left(\tilde{p}^e, \tilde{p}^b\right) \tag{9}$$

$$L_{fl} = \tau^2 \sum_{b=1}^{B} KL\left(\tilde{p}^{m(B)}, \tilde{p}^b\right) \tag{10}$$

where $L_{fl}$ and $L_{el}$ represent the KL divergence between each branch and the Fusion Logits and Ensemble Logits, respectively. Therefore, the total loss for the collaborative learning targets is:

$$L_{cl} = L_{fl} + L_{el} \tag{11}$$

## Teacher guidance

The pre-trained teacher can provide stable and comprehensive guidance, using softened logits to transfer knowledge to the student. Similar to Eq (9), the distillation losses for all branches with respect to the teacher are represented as:

$$L_{tg} = \tau^2 \sum_{b=1}^{B} KL\left(\tilde{p}^t, \widetilde{p}^b\right) \tag{12}$$

## Two-stage optimization and training

This study introduces a ramp-up weight [36] and designs a two-stage optimization strategy to control the weight of the distillation loss at different stages:

$$w(i) = e^{-5(1-i/E)^2} \tag{13}$$

where $E$ determines the duration of the weight change period, and $i$ is the $i$-th epoch with the rate of weight increase being smoothed through the empirical application of a factor of -5. During the training process, the initial parameter $a$ is utilized to regulate the epoch at which the ramp-up weight commences, with the setting range of $a$ being between 0 and 300. Finally, the overall loss function for THFKD is represented as:

$$L_{\text{total}} = \sum_{b=1}^{B} L_{ce}^b + w(i)\left(L_{cl} + L_{tg}\right) \tag{14}$$

where $L_{ce}^b$ is the cross-entropy loss between the $b$-th branch and the ground-truth label $y$. During the early stages of training, the parameters of the student model are not yet stabilized and are prone to being affected by noise. Therefore, the initial value of $w(i)$ is set to a relatively small magnitude, and the weight is gradually increased over a predetermined period to guide the student model's optimization smoothly. In Eq (14), the loss term $L_{tg}$ dominates, enabling the pre-trained teacher model to provide more comprehensive guidance in the initial phases. When $i = E$, the period ends, and the weight is set to 1. At this point, the student has gained a more comprehensive understanding, and diversifying the collaborative learning target becomes more important in the subsequent training. During training, the teacher model's parameters remain unchanged. Taking ResNet32 on CIFAR-100 as an example, the shape of the concatenated logits of all branches is $100 \times 4$ (where 4 is the number of branches $B$), and the loss for each branch is computed along the branch dimension. The performance of the model is primarily evaluated by classification accuracy(%), which represents the proportion of correctly classified samples to the total number of samples across multiple datasets.

## Experiments

This section first introduces the datasets, model architectures, and experimental settings used. Subsequently, we present the experimental results to evaluate the proposed THFKD method's performance comprehensively. In the ablation study, we analyze the impact of teacher guidance loss, collaborative learning loss, and the number of branches on the model's performance. We compare THFKD with other branch-based OKD methods to evaluate its effectiveness in alleviating the homogenization problem.

### Experimental setups

**Datasets and model architectures.** The THFKD method was evaluated on two widely used datasets: CIFAR-100 [37] and Tiny-ImageNet [38]. The CIFAR-100 dataset consists of 100 classes with a total of 60,000 RGB images of size $32 \times 32$, with the training set containing 50k images and the validation set containing 10k images. The Tiny-ImageNet dataset contains 200 classes with 20,000 RGB images of size $64 \times 64$. It includes 100,000 training images and both 10,000 testing and 10,000 validation images. All images were processed and normalized using channel-wise mean and standard deviation, and standard data augmentation techniques were applied. The experiments used various model architectures, including ResNet [1], WideResNet [39], MobileNet [40], and ShuffleNet [2].

 **Settings.** Optimization is performed using stochastic gradient descent with Nesterov momentum, with an initial learning rate set to 0.05 and momentum to 0.9 [29]. Training is conducted for 300 epochs with a batch size of 128. The learning rate is divided by 10 at epochs 150 and 225, weight decay is set to 5E-4, and the temperature $\tau$ is set to 3. The number of branches $B$ for the THFKD experiments is 4 [33]. All experiments are conducted using the PyTorch framework on an NVIDIA 4090 GPU.

### Comparison with KD methods

**Results on CIFAR-100.** The experiments on CIFAR-100 demonstrate that THFKD consistently outperforms cross-entropy-trained students by at least 3% across various architectures, as shown in Table 1. Compared to SOTA methods, THFKD achieves superior performance with identical teacher-student architectures. For instance, it surpasses ReviewKD by 1.52% with ResNet110-ResNet32 and by 0.66% with ResNet56-ResNet20. Different architectures

**Table 1. Accuracy (%) of KD methods on CIFAR-100.**

| Teacher | ResNet58 72.34 | ResNet110 74.31 | ResNet32x4 79.42 | WRN40-2 75.61 | WRN40-2 75.61 | ResNet32x4 79.42 | ResNet50 79.34 |
|---|---|---|---|---|---|---|---|
| Student | ResNet20 69.06 | ResNet32 71.14 | ResNet8x4 72.50 | WRN16-2 73.26 | WRN40-1 71.98 | Shufflenetv2 71.82 | Mobilenetv2 64.6 |
| KD [11] | 70.66 | 73.08 | 73.33 | 74.92 | 73.54 | 74.45 | 67.35 |
| FitNet [28] | 69.21 | 71.06 | 73.50 | 73.58 | 72.24 | 73.54 | 63.16 |
| RKD [22] | 69.61 | 71.82 | 71.90 | 73.35 | 72.22 | 73.21 | 64.43 |
| AT [26] | 70.55 | 72.31 | 73.44 | 74.08 | 72.77 | 72.73 | 58.58 |
| CRD [23] | 71.16 | 73.48 | 75.51 | 75.48 | 74.14 | 75.65 | 69.11 |
| OFD [24] | 70.98 | 73.23 | 74.95 | 75.24 | 74.33 | 76.82 | 69.04 |
| ReviewKD [29] | 71.89 | 73.89 | 75.63 | 76.12 | 75.09 | **77.78** | 69.89 |
| **THFKD** | **72.55** | **75.41** | **75.82** | **76.31** | **75.85** | 77.51 | **70.02** |

impact knowledge distillation performance. High-capacity model pairs, such as ResNet32x4-ResNet8x4 and WRN40-2-WRN16-2, are harder to improve due to overfitting with limited data. Using heterogeneous architectures, THFKD and ReviewKD achieved comparable performance on ResNet50-MobileNetV2 and ResNet32x4-ShuffleNetV2, limited by the incompatibility of ShuffleNet with the multi-branch architecture.

**Result on Tiny-ImageNet.** To further evaluate the generalization capability of the THFKD method, we conducted experiments on the large-scale Tiny-ImageNet dataset. As shown in Table 2, THFKD using ResNet34 as the teacher model and ResNet18 as the student model outperforms Vanilla KD by 1.98%. Compared to previous methods, THFKD surpasses the SOTA method NRKD by 0.71%, achieving the best performance. This demonstrates that the proposed method is also effective on large-scale datasets, and the diverse knowledge provided by the THFKD architecture can better enhance the student's generalization ability.

**Result on CIFAR100-LT.** A long-tailed dataset has a highly imbalanced sample distribution, where the number of samples in a few categories is significantly larger than that in most categories. As shown in Table 3, although THFKD is not explicitly designed for long-tail classification, it has demonstrated strong classification performance on the long-tailed dataset CIFAR100-LT. The heterogeneous branches designed in this paper allow each branch to specialize in different categories and data characteristics, thereby enhancing the model's ability to capture information from tail classes. Building on this architecture, our progressive feature fusion module enables effective integration of differentiated features, while the knowledge distillation mechanism facilitates knowledge sharing and complementarity specifically for tail categories. This indicates that the student model is capable of learning comprehensively across all categories, avoiding bias toward the more frequent classes in the early stages of training, and further optimizing the model with diversified knowledge to enhance its generalization ability, thereby improving the classification performance of the model on the tail categories.

## Ablation experiments

**Effect of different loss.** Different combinations of loss impact classification accuracy, as shown in Eq (11): $L_{cl} = L_{fl} + L_{el}$. As shown in Fig 4, training with the collaborative learning

**Table 2. Accuracy (%) of KD methods on Tiny-ImageNet. ResNet34-ResNet18.**

| Teacher | Student | KD [11] | AT [26] | CRD [23] | AFD [41] | FFKD [42] | NRKD [30] | **THFKD** |
|---|---|---|---|---|---|---|---|---|
| 66.84 | 65.14 | 67.66 | 67.76 | 67.94 | 68.10 | 68.51 | 68.93 | **69.64** |

**Table 3. Accuracy (%) on CIFAR100-LT with imbalance factor of 100 and 50. ResNet110-ResNet32.**

| Method | | LDAM Loss [43] | LADE [44] | RIDE [45] | NCL [46] | SHIKE [47] | **THFKD** |
|---|---|---|---|---|---|---|---|
| CIFAR100-LT | 100 | 44.4 | 45.4 | 48.3 | 54.2 | 56.3 | **58.1** |
| | 50 | 49.2 | 50.5 | 52.6 | 58.2 | 59.8 | **60.7** |

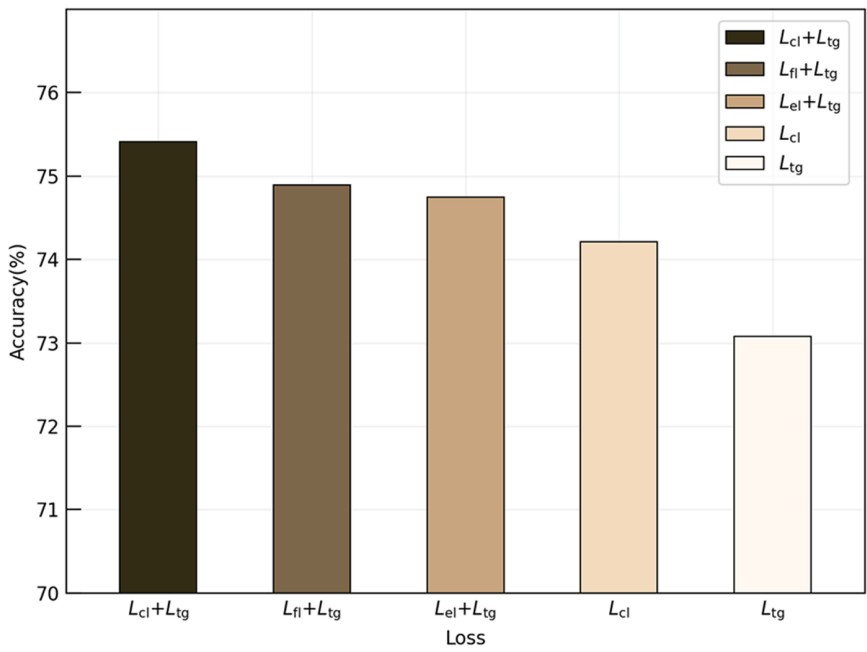

**Fig 4. Effect of different loss on accuracy.**

target ($L_{cl}$) loss yields a 1.17% improvement in classification accuracy compared to training with the teacher-guided ($L_{tg}$) loss. This indicates that the collaborative learning target provides diverse dynamic knowledge during training, enhancing the student's generalization ability. However, training with only static or dynamic knowledge performs less than joint training. Using $L_{fl} + L_{tg}$ and $L_{el} + L_{tg}$ losses leads to an improvement of approximately 0.4%. Ultimately, training with the proposed THFKD ($L_{cl} + L_{tg}$) approach through the two-stage optimization strategy provides appropriate knowledge to the student model at different stages, achieving the best performance.

**Effect of loss in different stages.** We analyze the impact of loss at different stages by visualizing the changes in accuracy and loss curves. As shown in Fig 5(a), during the early phase (the first 150 epochs), THFKD demonstrates a mean advantage in classification accuracy over the individual use of $L_{cl}$ and $L_{tg}$ for training, with an improvement ranging approximately from 2% to 5%. Furthermore, it can be observed that between epochs 90 and 110, the $L_{cl} + L_{tg}$ curve exhibits more minor fluctuations, with an amplitude of approximately 3%, whereas the fluctuations for $L_{cl}$ and $L_{tg}$ alone are around 12% and 10%, respectively. In Fig 5(b), although the loss value of $L_{tg} + L_{cl}$ is approximately four times that of $L_{cl}$, the loss value continues to decrease as training progresses, indicating that the teacher's guidance effectively directs the optimization trajectory of the student. Additionally, it can be observed that the $L_{tg}$ loss accounts for a larger proportion of the total loss compared to the $L_{cl}$ loss, suggesting that the

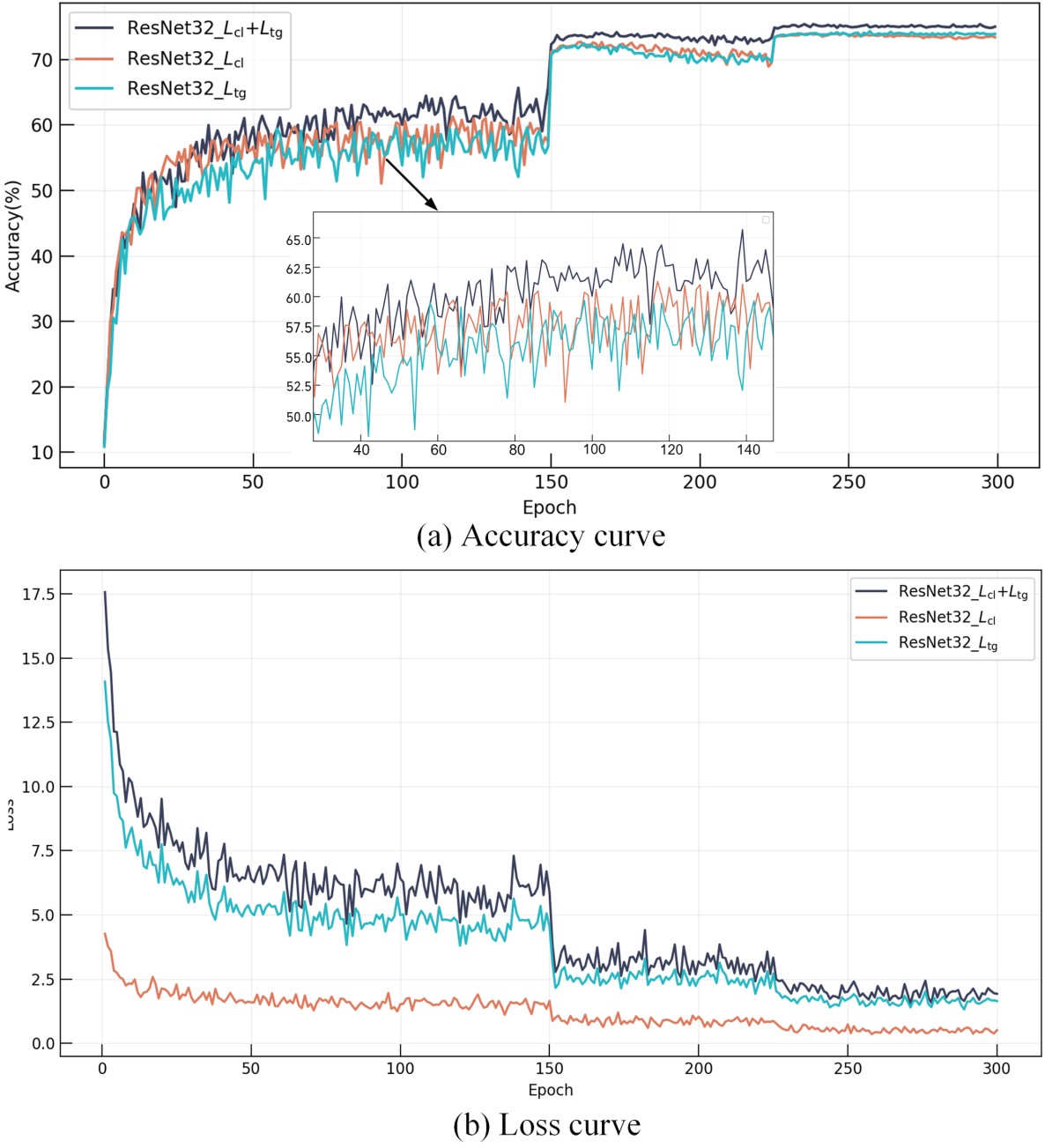

**Fig 5. Effect of loss in different stages.** (a) Accuracy curves with different losses. (b) Loss value change curves.

teacher's guidance dominates the early gradient updates of the student model. After the learning rate drops at epoch 150, entering the later training stage, the red accuracy curve ($L_{cl}$) in Fig 5(a) is slightly higher than the cyan one ($L_{tg}$), suggesting that diversified dynamic knowledge further optimizes the student model. However, a reversal occurs in the final stages, which can be attributed to the lack of guidance from the teacher in the early phases. This

absence leads to the accumulation of erroneous information within the collaborative learning objectives, and the impact of this phenomenon becomes further amplified as the training progresses into the later stages.

**Effect of two-stage optimization.** By controlling the period parameter $E$ and $a$ the starting parameter, we evaluated the impact of the weight increase period and the two-stage optimization strategy on the model performance. As shown in Table 4, when the period length is controlled during the early stage of training (before the learning rate decreases at epoch 150), the impact on model accuracy is minimal. However, when the period extends into the later stages of training, the change in weight loss is detrimental to the convergence of the student model, leading to a significant drop in accuracy. When $E$ is 0 or 300 (i.e., a single-stage optimization strategy), the loss weight gradually increases or remains constant throughout the training process, resulting in the classification accuracy of THFKD being lower than that obtained with the two-stage optimization strategy.

In the training phase, the starting parameter $a$ controls at which epoch THFKD begins to use ramp-up weights. As shown in Fig 6, when $a = 0$, the accuracy curve of the model steadily rises, indicating that THFKD completes the early learning phase within 80 epochs, laying the foundation for subsequent training, thereby maintaining the highest accuracy. However, when a is set to 30, 60, or 90, the accuracy sharply increases at epoch=$a$ (when the ramp-up weight is first applied) but then drops immediately, suggesting that changing the loss weight mid-training disrupts the optimization of the model. Therefore, by using ramp-up weight

**Table 4. Impact of the period parameter *E*. (Accuracy%)**

| E | 0 | 40 | 80 | **120(THFKD)** | 160 | 300 |
|---|---|---|---|---|---|---|
| Accuracy | 74.71 | 75.14 | 75.23 | **75.41** | 74.64 | 74.70 |

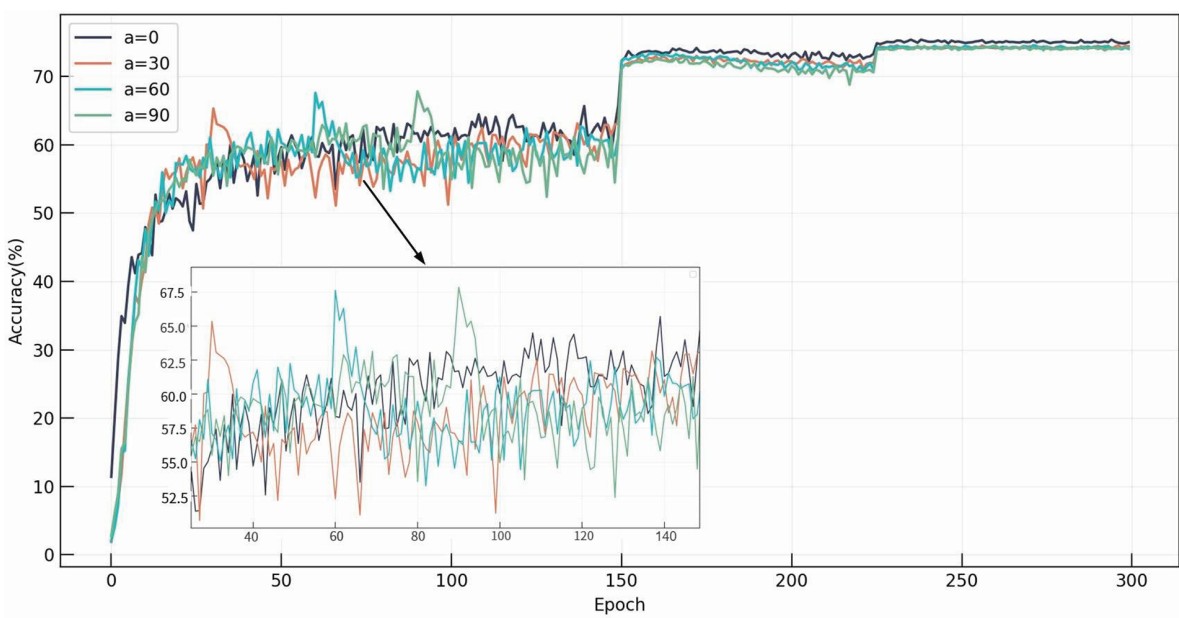

**Fig 6. Impact of the parameter *a*.**

at the beginning of training and setting the period length to 120, THFKD achieves the best two-stage optimization strategy.

**Impact of temperature parameter $\tau$.** The temperature parameter $\tau$ has a notable impact on the performance of the THFKD method. As shown in Table 5, the highest accuracy (75.41%) is achieved when $\tau = 3$, making it the optimal setting. A lower temperature ($\tau = 2$) may result in insufficient soft target information, limiting the learning capability of the student model. On the other hand, as the temperature increases to $\tau = 4$ and $\tau = 5$, the performance gradually declines, indicating that excessively high temperatures may overly smooth the probability distribution, leading to the loss of critical information. Therefore, $\tau = 3$ represents a balanced temperature that maintains sufficient information while avoiding excessive smoothing, providing effective guidance for the student model.

**Effect of branch number.** We set the total number of branches $B$ to range from 2 to 6 to analyze the impact of branch quantity on performance. As shown in Fig 7, THFKD exhibits a performance improvement of about 2% (from 73% to 75%) when the number of branches is increased from 1 to 4. However, further increasing the number of branches does not effectively improve performance. This is because the multi-branch architecture is independently constructed at higher layers. When the number of branches is increased, the complexity of the model increases, and the knowledge transfer becomes more chaotic. Thus, continuously increasing the number of branches does not sustain performance improvement. We observed comparable improvements in accuracy when increasing the number of branches from 3 to

**Table 5. Impact of temperature parameter $\tau$. (Accuracy%)**

| $\tau$ | 2 | 3 | 4 | 5 |
|---|---|---|---|---|
| THFKD | 74.29 | **75.41** | 74.93 | 74.72 |

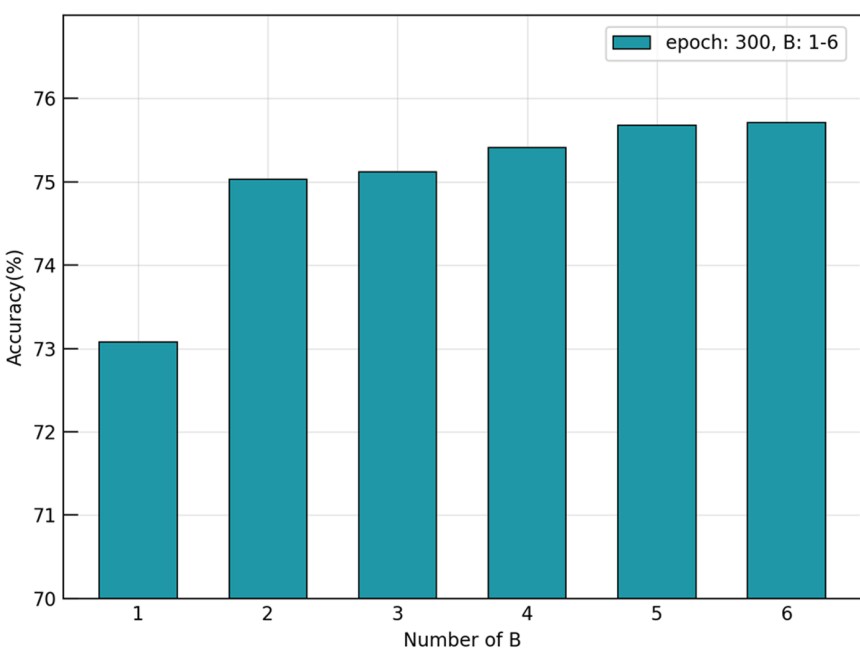

**Fig 7. Effect of branch number.**

4 and from 4 to 5. Although adding branches during testing does not incur additional costs, using 5 branches during training increases computational overhead, including higher GPU memory usage and longer training times, as the number of channels in the auxiliary blocks grows exponentially. Throughout the experiments, we balanced the performance gains against the computational costs and found that 4 branches were sufficient to achieve substantial performance improvements. Consequently, we selected 4 branches as the optimal configuration for our experimental setup.

**Homogenization problem.** We compared the online knowledge distillation methods based on multi-branch models, including CL [48], ONE [16], and OKDDip [32] (all configured with four branches during training), with AFID [49], which employs dual independent models. As shown in Table 6, the first row presents the baseline classification accuracies of the high-parameter models ResNet8x4 and WRN16-2, which are 72.50% and 73.26%, respectively, both higher than the 71.14% accuracy of ResNet32. However, when trained using CL, ONE, and OKDDip, the classification accuracies of these high-parameter models (ResNet8x4 and WRN16-2) decrease compared to ResNet32 (72.56% >61.81% and 72.43%; 73.50% >61.66% and 72.86%; 74.32% >72.75% and 73.96%). Notably, in the AFID, the independent model training approach does not exhibit such accuracy degradation (74.05% <74.56% and 74.94%). In contrast, THFKD shows a 1.09% improvement over OKDDip when using ResNet8x4 and a 1.37% improvement over AFID when using WRN16-2 while avoiding the phenomenon where high-parameter models (ResNet8x4 and WRN16-2) underperform ResNet32 in classification accuracy. This demonstrates that THFKD adapts to various model architectures and effectively mitigates the problem of homogenization.

**Diversity analysis.** By visualizing the Euclidean distance between the logits of each pair of branches and the ensemble prediction accuracy of the branches, we analyze the improvement in diversity. As shown in Fig 8, the Fig 8(a) compares the Euclidean distances of the four methods. It can be seen that after several rounds of optimization, the diversity rapidly increases, with THFKD maintaining stability, while OKDDip, ONE, and CL exhibit a downward trend. Our method is clearly superior to the other three methods throughout the training process. The Fig 8(b) shows the ensemble prediction accuracy of the four methods. Benefiting from the increased diversity, THFKD consistently outperforms the other methods in terms of ensemble accuracy, indicating that THFKD effectively enhances the diversity of dynamic knowledge, improving the model's generalization ability.

**Ensemble predictions.** Ensembling the predictions of multiple branches can lead to stronger performance. As shown in Table 7, we retain three branches for the ensemble output (THFKD-E). Benefiting from the increased diversity, THFKD-E achieves a more effective ensemble. When the number of parameters in the ensemble model is less than or equal to that of the teacher model, the ensemble's predictions achieve higher accuracy than those made by the teacher model.

**Table 6. Accuracy (%) of OKD on CIFAR-100.**

| Student | ResNet32 71.14 | ResNet8x4 72.50 | WRN16-2 73.26 |
|---|---|---|---|
| CL [48] | 73.86 | 62.30 | 72.97 |
| ONE [16] | 74.25 | 62.65 | 73.21 |
| OKDDip [32] | 75.08 | 73.30 | 75.29 |
| AFID [49] | 74.89 | 75.24 | 75.72 |
| **THFKD** | **75.41** | **75.82** | **76.31** |

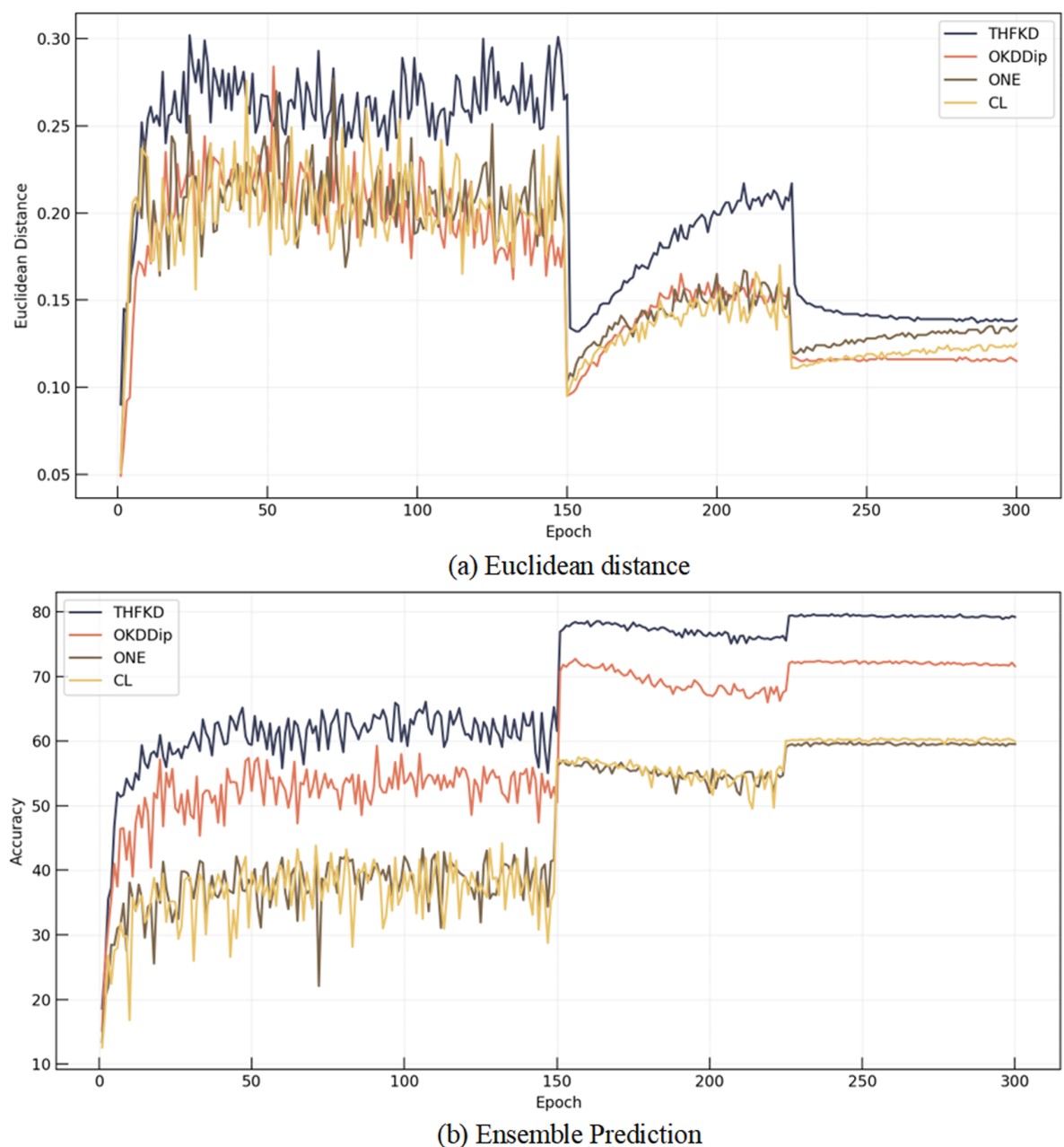

(a) Euclidean distance

(b) Ensemble Prediction

**Fig 8. Diversity comparison with ResNet8x4 as the student model.** (a): Euclidean distance. (b): Ensemble prediction.

**Table 7. Ensemble prediction Accuracy (%) and parameters (MB) of THFKD with three branches on CIFAR-100.**

| Teacher | ResNet56(0.86MB) 72.34 | ResNet110(1.71MB) 74.31 | ResNet32x4(7.43MB) 79.42 | WRN40-2(2.25MB) 75.61 | WRN40-2(2.25MB) 75.61 |
|---|---|---|---|---|---|
| Student | ResNet20 69.06 | ResNet32 71.14 | ResNet8x4 72.50 | WRN16-2 73.26 | WRN40-1 71.98 |
| THFKD | 72.76 (0.39MB) | 74.51 (0.58MB) | 75.12 (1.23MB) | 76.51 (0.70MB) | 75.24 (0.56MB) |
| THFKD-E | 76.29 (0.85MB) | 78.05 (1.40MB) | 80.21 (6.46MB) | 79.31 (2.62MB) | 78.33 (1.64MB) |

## Discussion

**Branches with different abilities.** THFKD's performance in the 300th epoch is shown under three model architectures. As shown in Table 8, the performance of the four branches gradually improves as the number of parameters increases (with the channel count in the auxiliary blocks growing multiplicatively), demonstrating differences in abilities. This provides a foundation for diversifying collaborative learning targets, allowing THFKD to meet varying precision requirements and enabling more flexible deployment.

To analyze the impact of heterogeneous branches in terms of width and depth, we conducted experiments on the ResNet32 model by varying the number of auxiliary blocks (NA) and their channel size (CA). Specifically, NA was set to 0, 1, 2, where NA = 0 indicates that each branch has the same structure without auxiliary blocks, and NA = 1 corresponds to the original experimental setting of THFKD. CA was set to either 1× or 2× the number of channels in the high-level layers (CA = $C \times 1$, $C \times 2$).

As shown in Table 9, although the absolute differences in accuracy are relatively small (mostly within 1%), some consistent trends can still be observed. Firstly, when NA=0, all branches perform poorly (ranging from 74.3% to 74.6%), which may be due to the lack of auxiliary structures limiting feature diversity. Introducing auxiliary blocks (NA=2) generally improves performance (e.g., Branch-3 reaches up to 75.64%), but the trend is not strictly monotonic—overly complex branches such as Branch-4 show a slight performance drop, possibly due to increased optimization difficulty. This suggests that simply stacking more modules does not guarantee better results, and moderate structural diversity may be more beneficial.

Impact of the number of channels (CA): Doubling the number of channels improves performance in some branches (e.g., Branch-2 and Branch-3 reach 75.77% and 75.90%, respectively), indicating that selective capacity expansion can enhance feature learning ability. However, for CA=$C \times 1$, $C \times 2$, the accuracy fluctuations of Branches 2–4 are relatively small (with the maximum and minimum accuracy differences being 0.45% and 0.16%, respectively), which suggests that uniform channel expansion (i.e., using identical auxiliary blocks) leads to similar capabilities among branches, thereby reducing the diversity of dynamic knowledge. The best performance is observed under the THFKD configuration (NA=1, CA=C×1), with Branch-1 achieving an accuracy of 75.03%. Although the performance improvement is modest, the accuracy variation among Branches 2–4 is relatively large (the maximum difference of

**Table 8. Accuracy (%) of branches with different abilities on CIFAR-100.**

| Student | Branch-1 | Branch-2 | Branch-3 | Branch-4 |
|---|---|---|---|---|
| Resnet32 | 75.03 | 75.36 | 76.20 | 76.61 |
| Resnet8x4 | 75.28 | 78.42 | 79.13 | 80.00 |
| WRN16-2 | 76.13 | 77.16 | 78.17 | 78.67 |

**Table 9. Accuracy (%) of heterogeneous branches at different depths and widths.**

| B | Branch-1 | Branch-2 | Branch-3 | Branch-4 |
|---|---|---|---|---|
| NA=0 | 74.36 | 74.49 | 74.33 | 74.65 |
| NA=2 | 74.83 | 75.02 | 75.64 | 75.06 |
| CA=$C \times 1$ | 74.51 | 74.71 | 75.16 | 75.15 |
| CA=$C \times 2$ | 74.44 | 75.77 | 75.90 | 75.74 |
| THFKD(NA=1) | 75.03 | 75.36 | 75.20 | 76.61 |

1.41% is observed between Branch 3 and Branch 4), indicating that well-designed heterogeneous branches in both depth and width help promote diversity in feature representation.

**Feature visualization.** This study uses Grad-CAM [50] to visualize the feature regions attended to by the model, as shown in Fig 9, including the heatmaps of each branch and PFF-3, along with the ground truth labels and predicted classes. Columns 1 and 2 show the cases

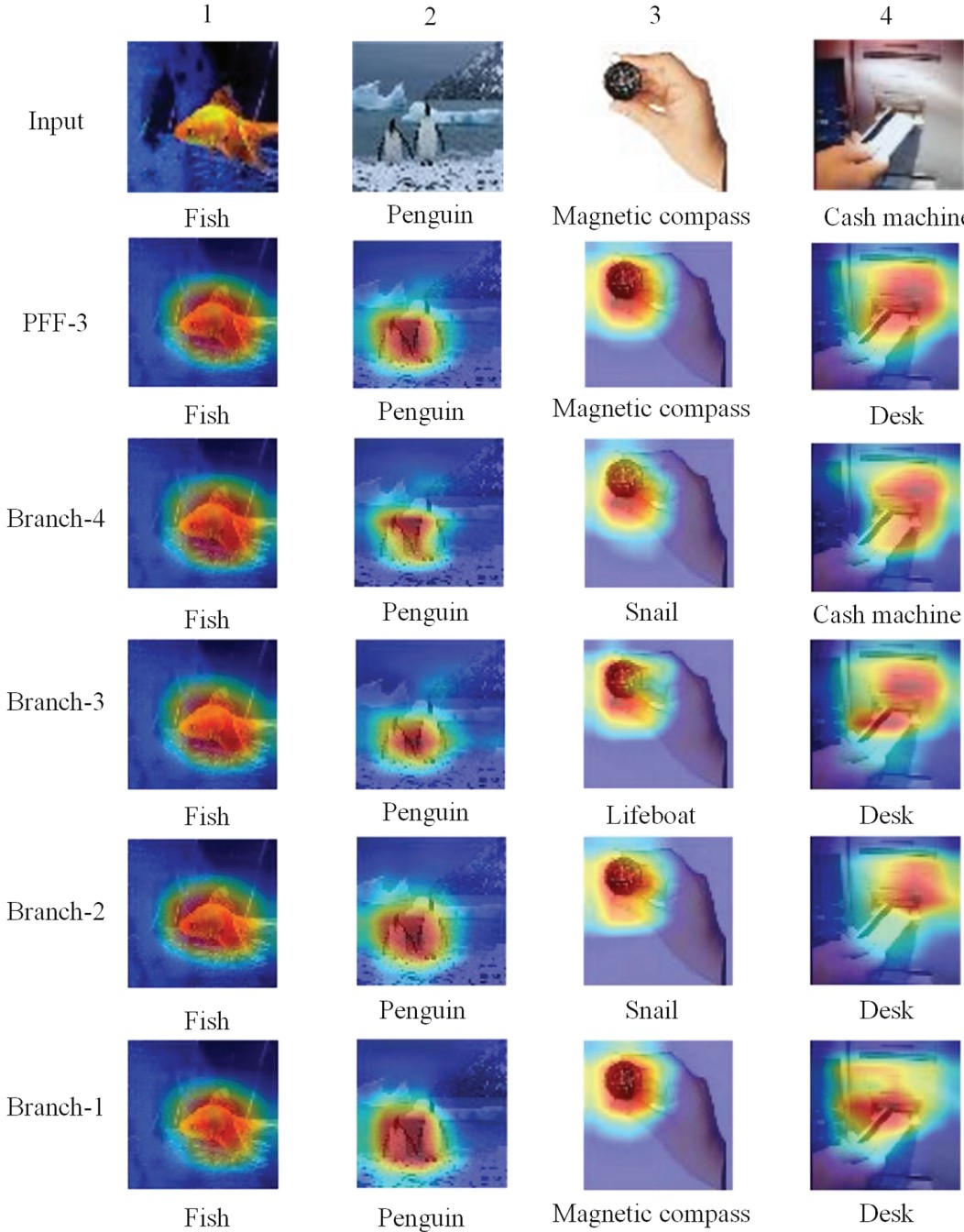

**Fig 9. Fusion module and the Grad-CAM for each branch are visualized in the Tiny-ImageNet dataset.**

where each branch and PFF-3 classify correctly, demonstrating high certainty for simple images. In Column 2, the larger-capacity branches focus more on local features. In Column 3, although each branch and PFF-3 focus on the correct regions, they predict three different categories. This demonstrates that the student can learn more comprehensively by softening logits to transfer similar information between categories and combining both static and dynamic knowledge. Only Branch-4 can correctly predict the class in the fourth column, demonstrating that THFKD can further optimize the student model in the later stages. We also observe that the feature heatmap of PFF-3, compared to those of Branches 1-4, captures more accurate feature information, indicating that PFF effectively fuses the differentiated feature representations.

## Conclusion

We propose a novel Two-stage optimization based on heterogeneous branch fusion for knowledge distillation (THFKD). In traditional knowledge distillation, the student model is solely guided by the fixed knowledge of the teacher model, which cannot supplement or expand upon this knowledge, thereby limiting the student's generalization capability. In contrast, our student model leverages heterogeneous branches and a Progressive Feature Fusion (PFF) module to generate collaborative learning objectives, enabling the transfer of diversified, dynamic knowledge rather than relying exclusively on the teacher model. During training, the dynamic knowledge is combined with the static knowledge from the pre-trained teacher, while the two-stage optimization strategy provides appropriate knowledge at different training stages, enhancing the student model's learning efficiency and generalization ability. Although no tests of statistical significance were carried out, our experimental results on standard datasets (CIFAR-100, Tiny-ImageNet) and long-tail datasets (CIFAR100-LT) suggest that THFKD may slightly improve the student model's classification accuracy and generalization ability. This demonstrates its broad application potential in image classification tasks. For instance, using ResNet110-ResNet32 on the CIFAR-100 dataset, the accuracy reaches 75.41%, a 1.52% improvement over the state-of-the-art (SOTA). Additionally, we provide an analysis and visualization to further understand the underlying mechanisms of the proposed method. However, our method also exhibits certain limitations. THFKD performs slightly below existing approaches on lightweight models such as ShuffleNet, and the range of model combinations utilized is not yet comprehensive. In future work, we aim to investigate these limitations further and extend THFKD to other domains, such as object detection and image segmentation, to explore its broader applicability.

## Author contributions

**Conceptualization:** Gang Li, Pengfei Lv, Yang Zhang.

**Formal analysis:** Zihan Ruan.

**Methodology:** Pengfei Lv, Yang Zhang, Chuanyun Xu.

**Supervision:** Gang Li, Zheng Zhou.

**Validation:** Ru Wang.

**Writing – original draft:** Gang Li, Pengfei Lv.

**Writing – review & editing:** Xinyu Fan, Pan He.

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
