## [Decision Letter · Decision Letter 0]

5 Mar 2025

PONE-D-25-00345Two-stage optimization based on heterogeneous branch fusion for knowledge distillationPLOS ONE

Dear Dr. Zhang,

Thank you for submitting your manuscript to PLOS ONE. After careful consideration, we feel that it has merit but does not fully meet PLOS ONE’s publication criteria as it currently stands. Therefore, we invite you to submit a revised version of the manuscript that addresses the points raised during the review process.

The reviewers raised comments that need to be addressed.

We look forward to receiving your revised manuscript.

Kind regards,

Alberto Marchisio

Academic Editor

PLOS ONE

Journal Requirements:

Reviewers' comments:

Reviewer's Responses to Questions

**Comments to the Author**

1. Is the manuscript technically sound, and do the data support the conclusions?

Reviewer #1: Yes

Reviewer #2: Yes

Reviewer #3: Yes

Reviewer #4: Yes

Reviewer #5: Yes

Reviewer #6: Yes

2. Has the statistical analysis been performed appropriately and rigorously? 

Reviewer #1: Yes

Reviewer #2: Yes

Reviewer #3: N/A

Reviewer #4: Yes

Reviewer #5: No

Reviewer #6: N/A

3. Have the authors made all data underlying the findings in their manuscript fully available?

Reviewer #1: Yes

Reviewer #2: Yes

Reviewer #3: Yes

Reviewer #4: Yes

Reviewer #5: Yes

Reviewer #6: Yes

4. Is the manuscript presented in an intelligible fashion and written in standard English?

Reviewer #1: Yes

Reviewer #2: Yes

Reviewer #3: Yes

Reviewer #4: Yes

Reviewer #5: Yes

Reviewer #6: Yes

5. Review Comments to the Author

Reviewer #1: The main content of this paper is focused on a two-stage optimization approach based on heterogeneous branch fusion for knowledge distillation. The authors propose a collaborative learning target that consists of Fusion Logits obtained from fused features and Ensemble Logits derived from the integration of branch logits. They introduce a multi-branch architecture where each branch shares parameters in the low-level layers but has independent parameters in the high-level layers. The authors also design a progressive feature fusion module to enhance the utilization of features and improve the representational capability of the model. Experimental results on standard datasets such as CIFAR-100, Tiny-ImageNet, and CIFAR100-LT demonstrate that the proposed method significantly improves the classification accuracy and generalization ability of the student model.

However, there are some issues that need improvement:

1. The limitations of existing methods are suggested to be refined to highlight the value of the proposed method. Related work needs to be further organized rather than just listed.

2. The contributions of the paper should be stated more clearly and explicitly. The innovation points and contribution statements at the end of the introduction section are suggested to be further refined and condensed to highlight the core values of this paper.

3. More high-quality papers need to be added to further highlight the value of the proposed method, such as the introduction or related work chapters, including but not limited to the following papers:

[1] Knowledge distillation: A survey[J]. International Journal of Computer Vision, 2021, 129(6): 1789-1819.

[2] Learning from Human Educational Wisdom: A Student-Centered Knowledge Distillation Method[J]. IEEE Transactions on Pattern Analysis and Machine Intelligence, 2024, 46(6): 4188-4205.

[3] Improving Knowledge Distillation via Head and Tail Categories[J]. IEEE Transactions on Circuits and Systems for Video Technology, 2024, 34(5): 3465-3480.

and so on…

4. https://github.com/GitLpf/THFKD. The github URL provided by the author is missing the necessary code and other information. However, one of the most valuable contributions of this paper is the availability of their open-source code. Therefore, it is strongly recommended that the authors carefully organize the relevant de-sensitization data, code, and processing flow introduction into github or other open-source forms, and give them in the form of URL links in the Abstract, so as to enhance the influence of the text work and attract more interested readers to follow the work of this paper.

5. In the Section of Methodology, the two-stage optimization strategy and progressive feature fusion module should be described in more detail with mathematical formulations.

6. In the Section of Methodology, the loss functions and training procedure should be explained more thoroughly.

7. In the Section of Experiments, The experimental setup (datasets, model architectures, hyperparameters) should be described in more detail.

8. More ablation studies should be added to analyze the impact of different components (e.g. fusion module, loss weights). Moreover, more tables are suggested to be added to visually present the comparison results of the performance before and after ablation.

9. The results should be analyzed in more depth, with insights into why the proposed method works well.

10. The results should be analyzed in more depth, with insights into why the proposed method works well.

11. In the Section of Conclusion, the conclusion should summarize the key findings and contributions. Limitations of the current work and potential future directions should be discussed.

12. The writing should be polished for clarity, grammar, and flow. Some sentences are overly long and could be broken up. Consistent terminology should be used throughout (e.g. "student model" vs "target model").

Reviewer #2: In this paper, the authors investigate knowledge distillation methods and propose a two-stage optimization based on heterogeneous branch fusion for knowledge distillation. The paper is interesting and addresses intriguing questions. The authors conduct experiments rigorously and draw conclusions based on the presented data. While their analysis is thorough and the paper is generally presented in an intelligible manner, it contains ambiguities that hinder readability and discourage further engagement.

begin{itemize}

item

The authors present nine figures in this paper. Unfortunately, the way these figures are presented significantly hinders the readability of the paper. The captions, without the corresponding figures, are scattered throughout the paper, preventing readers from reading the captions while simultaneously viewing the corresponding figures. All nine figures related to these captions are located at the very end of the paper. Moreover, the figure corresponding to the caption ‘Fig 1. Two training frameworks. (a) using the OKD method and (b) using the KD method.’ is placed as the eighth figure in the list of nine figures located at the end of the paper. Thus, the order of the captions throughout the paper is completely different from the order of the figures at the end. Additionally, the eighth figure in this list contains two parts: ‘(a) Multi-branch OKD’ and ‘(a) Multi-branch KD’, with each part labeled as (a), which is inconsistent with the caption for Figure 1. This should be corrected. Furthermore, each figure should be positioned alongside its caption so that readers can read the figure descriptions while simultaneously viewing the corresponding figures. This will enable readers to easily correlate, for example, the notation used in the captions with the content of the figures. In the current setup, this process is severely hindered. To facilitate reading and encourage engagement with the paper, the authors should reorganize the way their figures are presented.

The notation used in equations (1), (2), and (3) should be explained in more detail. For example, the authors should clarify the relationship between z^b^ and its components zkb and zib by defining zb=[z1b,dots,zkb]. Similarly, it would be helpful to define y=[y1,dots,yk].

The definitions of pkt and pks labeled by (3) are identical. To avoid repetition, the authors should define

p˜kj=expzkj/τ∑i=1Kexpzij/τ,  j∈t,s.

displaystyle tilde{p}_k^j = displaystyle frac{exp(z^j_k/tau)}{ sum_{i=1}^{K}exp(z^j_i/tau)}, quad jin {t,s}.

item

Examples of the fusion function $D$ used in (6) should be provided. Alternatively, references to the literature where such examples are discussed should be included.

item

In (6), ``$bin 2,dots ,B$" should be replaced with $bin { 2,dots ,B }$.

end{itemize}

Reviewer #3: General comments:

The authors identify a core problem with offline knowledge distillation: that it relies solely on the fixed knowledge of the teacher, which may limit the student’s ability to generalize. Also, the authors identify a limitation of selected online knowledge distillation techniques, where students learn concurrently with teachers or with multiple branches: the teacher or branches may not provide sufficient guidance at the early stages of training.

They offer a novel two-stage solution with the same architecture in both stages, but where the weights of different loss functions and learning rates change between stages. During the first stage a fixed teacher model offers more guidance, the relative weight of which diminishes during this stage. The second stage simultaneously combines, with equal weights, the fixed teacher’s guidance and dynamic information from heterogeneous branches. The heterogeneous branches are combined in two different ways: ensemble logits and progressive feature fusion.

Results are presented demonstrating improvements in accuracy over state-of-the-art knowledge distillation models using the same teacher-student architectures.

My main question is in relation to lines 223-235. The text cites Eq. 14, which doesn’t exist in this paper, and discusses Eq. 12. It is stated that “In the early stages of training, the student model is susceptible to noise, and the teacher-guided loss term L_{tg} in Eq. 14 dominates. Therefore, the initial weight w(i) is set to a small value and gradually increased over a certain period to allow the teacher to guide the student more smoothly, thereby reducing the impact of noise.” However, in the equation both L_{cl} (the collaborative learning loss) and L_{tg} are multiplied by the weight. If multiplying L_{tg} by w(i) was a typographical error, then the article is generally consistent. However, if the equation was written correctly, the authors need to justify their claim that the teacher-guidance dominates at the beginning of training, because this is the whole point of the article. This doubt led to my publication recommendation.

Data availability:

The data are publicly available.

Comments about exposition of the model:

The model is generally well described in well written English.

In the discussion of progressive feature fusion, on line 198, it is not clear if “w” is a scalar or vector. The caption of Figure 3 suggests it’s a vector. Please clarify this in the text.

Also, the parameter "a" is not introduced during the presentation of the model.

Comments about interpretation of results:

The methodology is clearly explained. Results are presented under many different situations and, in general, are well interpreted (the results support the interpretation/discussion). I offer some specific comments and questions below.

Lines 305-306: “significantly” should be changed for a word without statistical meaning, perhaps “generally outperforms by an average margin of....”. Also, characterize the fluctuations statistically and compare the values.

Lines 308-310: There is an intermediate period where the red line is above the blue line, but this inverts by the end of the period.

Lines 334-336 (Figure 7): There seems to be an equivalent increase in accuracy from 3 to 4 branches and 4 to 5 branches. Why not use 5 branches?

Lines 341-354: Rewrite the text to make it clear which column is being analyzed. Your statements seem general but are only true in one of the columns.

Lines 375-379: Is it surprising that the branches with more complex architectures (because of the increasingly complex auxiliary blocks) have higher accuracies?

Grammar and Typological Errors

In general, the text is well written and easily understood. In relation to grammar and typographical errors, I list the following observations.

Figure 1 labels both panels as 1a.

Line 161: I suggest writing something like, “to the target branch structure” instead of “to the target branch”?

Line 226: “Equation (14)” should be “Equation (12)”.

Line 248: The description of the number of images in Tiny-ImageNet is a little confusing. Try to explain this part more clearly.

Line 292: Wrong equation reference.

Figure 5: “Loss cure” should be “Loss curve”.

Line 313: Correct the English.

Line 372: Clarify this sentence.

Figure 9: “Branch”, not “Branach”.

Line 393: Clarify this sentence.

Line 412: “further to” -> “to further”.

Reviewer #4: The manuscript proposes a novel two-stage optimization strategy for knowledge distillation, integrating static knowledge from a teacher model with dynamically acquired knowledge through heterogeneous branches. The idea is innovative, and the empirical results show its effectiveness across various datasets. However, the following areas require improvement:

Theoretical Justification:

The heterogeneous branch structure and progressive feature fusion module are central to the proposed approach, yet the theoretical foundation behind their effectiveness is not fully explored.

A deeper mathematical discussion on how heterogeneous branches enhance student model learning would strengthen the study.

Ablation Studies:

While the experiments are comprehensive, further analysis of branch heterogeneity (e.g., varying branch widths, depths) is needed to isolate its impact.

The study should also evaluate the model's performance with and without the PFF module to clarify its contribution.

Extending experiments to additional tasks (e.g., object detection, segmentation) would help assess generalization beyond classification.

Hyperparameter Sensitivity:

A sensitivity analysis on the ramp-up weight period (E) and temperature parameter (τ) would be valuable.

Results with different teacher architectures (e.g., ViTs, EfficientNets) could offer additional insights.

Clarity and Presentation:

The figures should be better annotated, particularly Figures 2 (multi-branch structure) and 3 (PFF module).

Some terminology is used interchangeably (e.g., "auxiliary blocks" vs. "auxiliary branches"), which could be standardized for better readability.

Comparison with Related Work:

The study discusses knowledge distillation and online distillation but does not compare its approach with recent hybrid distillation methods (e.g., semi-online KD, ensemble-based KD).

The authors should clarify how their method differs from or improves upon Snapshot Distillation and Focal Distillation.

Overall, the paper presents a well-formulated approach with strong empirical validation. If the authors address the above concerns, the manuscript will be significantly strengthened.

Reviewer #5: I indicated that the statistical analysis has not been performed appropriately because the authors refer several times to a "significant difference" or "significant improvement" but no proper statistical analysis justifying such significance is performed. Following are the places of the manuscript where the word "significant" is used:

1) Abstract, Lines 70-71, Lines 405-406: "THFKD significantly improves the student model's classification accuracy".

2) Lines 56-57: "Experimental results show that the integration of multiple branches can significantly improve performance."

3) Lines 266-267: "Different architectures significantly impact knowledge distillation performance."

4) Line 291: "Different combinations of loss significantly impact classification accuracy".

5) Lines 305-306: "THFKD(Lcl + Ltg) significantly outperforms the accuracy of models trained with Lcl or 305 Ltg individually".

6) Lines 334-335: "As shown in Fig7, THFKD exhibits significant performance improvement when the number of branches is set to 4."

7) Line 338: "When the number of branches is significantly increased".  remove the word "significantly" here as the number of branches takes the values 1, 2, 3, 4, 5.

8) Line 348: "using the CL, ONE, and OKDDip methods significantly decreased".

My recommendation is to either:

a) Provide a statistical analysis proving that the differences and improvements mentioned are indeed significant, e.g. by computing confidence intervals for the plotted curves and bars.

b) Defining precisely what is meant by "significant". Perhaps it is given by a change in the number of correctly classified images above a pre-defined threshold?

c) Remove the word "significant" from all the aforementioned sentences.

My additional comments on the manuscript follow, grouped into GENERAL, MAJOR, and MINOR comments.

GENERAL COMMENTS: (Total = 4)

G1) Please make all vertical axes start at 0 in ALL plots, in order to avoid a misinterpretation of the magnitude of the differences among the methods compared. This is specially apparent in Figures 4 and 7.

G2) Please define how model performance is measured.

G3) Please clarify what is shown in the tables. Instead of using "Results of..." use e.g. "Accuracy on test sets of...".

G4) Please define "capacity". E.g. in "high-capacity models" which is used a couple of times.

MAJOR COMMENTS: (Total = 9)

M1) The definition of the cross-entropy loss should be corrected. Currently:

a) the summation should be over the samples, in addition to the classes.

b) the label "y" is not fully defined for a correct interpretation of the cross-entropy loss. In fact, labels can be anything, they don't even need to be numbers. Please consider mentioning the one-hot encoding of the labels or the use of the indicator function applied on "y(i)" of the true label "k" of each sample "i".

c) a minus sign is missing, which states that the objective is to minimize the loss, not maximize it (in fact, the loss should achieve its minimum of zero when the estimated probability for a sample in class k of being in class k is equal to 1.0).

M2) Fig. 3:

a) The fused features from the previous level is called F^m_{b-1} in the text and F^z_{b-1} in the figure. Please unify notation. Also, what is `m`?

b) Please indicate that BN = Batch Normalization and ReLU is "Rectified Linear Unit" (e.g. as a footnote).

M3) Line 269 reads "...THFKD improves by 0.27% with ResNet50-MobileNetV2...", but:

a) the value 0.27% is not correct as the difference on the rightmost column of 70.02 - 69.89 = 0.23;

b) the difference between THFKD and ReviewKD for ResNet32x4-ShuffleNetV2 is 77.51 - 77.78 = -0.27.

Hence, I don't find it fair to write "THFKD improves by ... with ResNet50-MobileNetV2" while writing "achieves comparable performance" when talking about the ResNet32x4-ShuffleNetV2 architecture, as their differences are about the same but with different sign (0.23 vs. -0.27, respectively).

Therefore, I encourage the authors to write "THFKD and ReviewKD achieve comparable performance with ResNet50-MobileNetV2 and ResNet32x4-ShuffleNetV2".

M4) Lines 254-259: Please provide a justification for the selection of the different hyperparameters, or a reference that suggests those choices.

M5) Line 312: "Effect of two-stage optimization"

Consider improving this subsection. The following points make the description unclear:

a) The starting parameter "a" is mentioned in the first paragraph but only defined in the second paragraph.

b) The starting parameter "a" is sometimes written in italics (correct) and sometime writen in regular typescript (incorrect, as it gets confused with the undetermined English article "a").

c) The first paragraph starts "By controlling the period parameter E and a the starting parameter a, ...", i.e. "a" is incorrectly used twice, generating confusion.

d) In the second paragraph we read the following phrase: "the accuracy sharply increases in the current epoch". What is "current epoch"? It would be clearer if you wrote: "the accuracy sharply increases

at the epoch 'a' when the ramp-up weight starts being used".

e) Fig. 6's caption should read rather: "Impact of parameter 'a'".

M6) Lines 406-410: The sentences describing the experimental results should be improved. Specifically:

a) What is the "with ResNet32 and ResNet100" case? Perhaps you meant the "ResNet100-ResNet32" case corresponding to column 2 in Table 1?

b) It would help to have the table numbers from which the results are taken for being able to easily fetch the results and understand better (e.g. "see column 2 in Table 1").

c) Reporting the absolute accuracy doesn't help much in understanding the benefits of the proposed method. The change in accuracy should also be reported w.r.t. to what is considered to be the SOTA method.

d) Why are those results selected for reporting in the Conclusions? Is it because the ResNet32 is considered to be a commonly used architecture? If this is the case, please indicate so.

e) Why is the THFKD with accuracy 58.1% chosen for reporting instead of the case with accuracy 60.7% shown in Table 3?

f) The final sentence should read "...we provide an analysis and visualization to further understand..." instead of "...further to...".

MINOR COMMENTS: (Total = 19)

m1) To simplify reading and avoid having to refer to the references section every time the reader encounters an acronym, please mention the full name of the acronym the first time the acronym is used. This happens several times (~10) in the manuscript (e.g. DML, ONE, CKA, FFL, AFF, etc.)

m2) Lines 69-74: I don't think this should be listed as contribution. On one hand, the first paragraph describes the experimental results, and on the other hand the use of accuracy and loss curves, and heatmaps are all pre-existing techniques for analyzing model's performance and understanding how the model works (e.g. heatmaps are done using the Grad-CAM method described in reference [46]).

m3) Lines 105-109: The authors indicate that "... the student model is susceptible to noise in the early stages of training...". However, they do not provide an alternative to tackle this problem, or at least it is not clear to me. If they provide an alternative, please better clarify it in the manuscript. Note that the authors do propose alternatives to SOTA methods for the other two subsections presented in "Related work", namely: "Knowledge distillation" and "Multi-branch architectures and feature fusion".

m4) Lines 145-150: I suggest improving the namings because:

a) the loss in (5) is referred to as "knowledge distillation";

b) the first paragraph in line 148 refers names "distillation loss" the K-L divergence loss L_{kl}. However, my first thought was that it is referring to L_{kd} because its name is "knowledge distillation" and L_{kl} is simply referred to as "loss" on line 146;

c) the loss in (2) is called "cross-entropy loss" therein, but on line 149 is called "label loss".

My suggested workaround is:

a) use "distillation loss" instead of just "loss" on line 146;

b) use "cross-entropy loss" instead of "label loss" on line 148.

m5) Fig. 2: It is not clear what the "high-level layers" are. Perhaps the position of the label "high-level layers" should be improved to clarify this.

m6) In expression (12) we read L_{cl} in the second term but it should be L_{el}.

m7) Following expression (12) we read "Eq. 14", but I it seems it should be "Eq. 12".

m8) Following expression (12) we read "When E = i, ...". Since "E" is a constant and "i" is the variable (the epoch), it is more natural and intuitive to write "When i = E, ...".

m9) Line 233: Why is the first dimension of the concatenated logits 128? Please justify.

m10) Lines 258: It seems the weight decay should be 5E-4, not "5x10 - 4" as is currently written.

m11) Table 1: How are methods in the rows and architectures in the columns sorted? By performance? By invention year? I suggest using a sensible criterion for sorting both rows and columns and indicate that criterion in the caption.

m12) Lines 291-292: what does it mean "Eq. 11: cl=fl+el." in the sentence "Different combinations of loss significantly impact classification accuracy, as shown in Eq. 11: cl=fl+el."?

m13) Line 347: A `%` sign is missing after 71.14.

m14) Lines 347-349: The statement seems to be partially incorrect as OKDDip's performance doesn't fall below the ResNet32 performance of 71.14%, according to Table 5.

m15) Fig 5:

a) Please correct the x labels to read "Accuracy curve" and "Loss curve".

b) Why is the ResNet32-L_{tg} Loss curve not plotted as is the case for the "Accuracy curve"?

m16) Line 372: "When the parameter is less than or equivalent to the teacher model". What does this mean? And what parameter are we talking about? Perhaps you meant "number of parameters" instead of "parameter"?

m17) Table 6: What does "MB" stand for in "parameters(MB)"?

m18) Lines 375-380:

a) The title says "different capacities" and the second sentence says "different capabilities". Is this correct or should "capabilities" be "capacities"?

b) What does it mean "the performance of the four branches gradually improves"? It improves with what? Perhaps you meant "Performance gradually improves as the number of branches improves"?

m19) Fig. 8: "Branch" is spelled incorrectly as "Branach".

Reviewer #6: • The cited research papers are mostly older than the year 2022. Hence, there is a space for comparison with the recent literature.

• There are some recent research articles related to the subject area covered in this research, like:

1. Gou, Jianping, Yu Chen, Baosheng Yu, Jinhua Liu, Lan Du, Shaohua Wan, and Zhang Yi. "Reciprocal teacher-student learning via forward and feedback knowledge distillation." IEEE transactions on multimedia (2024).

2. Guo, Wei, Xiang Li, and Ziqian Shen. "A lightweight residual network based on improved knowledge transfer and quantized distillation for cross-domain fault diagnosis of rolling bearings." Expert Systems with Applications 245 (2024): 123083.

3. Zhang, Sha, Jiajun Deng, Lei Bai, Houqiang Li, Wanli Ouyang, and Yanyong Zhang. "Hvdistill: Transferring knowledge from images to point clouds via unsupervised hybrid-view distillation." International Journal of Computer Vision 132, no. 7 (2024): 2585-2599.

• The reported accuracy of 69.64% should also be compared to the mentioned recent state-of-the-art (SOTA) systems.

• The accuracy is reported as the only performance metric in the manuscript. Other performance parameters like training/testing time may also be reported and compared with recent SOTA.

• More graphs may be included in the manuscript explaining the behavior of Performance Metrics for the proposed system.

• Conclusions should precisely address the claimed contributions made at the end of the Introduction section.

• English language mistakes may be checked using human experts or specialized language software.

6. PLOS authors have the option to publish the peer review history of their article (what does this mean?). If published, this will include your full peer review and any attached files.

Reviewer #1: No

Reviewer #2: No

Reviewer #3: No

Reviewer #4: No

Reviewer #5: **Yes: **Daniel Mastropietro

Reviewer #6: No

---

## [Author Response · Author response to Decision Letter 1]

26 Mar 2025

Dear Editors and Reviewers:

Thank you for your letter and for the reviewers’ comments concerning our manuscript. Those comments are valuable and very helpful for revising and improving our paper, as well as the guiding significance to our studies. We have carefully considered the comments and have made revision which is hoped to meet with approval. In order to highlight the changes that we have made, we additionally submit a marked version of the paper and revised portion which is marked in blue.

Reviewer #1:

1.The limitations of existing methods are suggested to be refined to highlight the value of the proposed method. Related work needs to be further organized rather than just listed.

Author response:

Thank you for your review and valuable suggestions. We completely agree with your viewpoint that the original manuscript's analysis of the limitations of existing methods was not sufficiently in-depth, and the organization of related work was somewhat loose. In the revised version, we have conducted a more in-depth analysis of the limitations of existing methods in the related work section and optimized the text accordingly.

2.The contributions of the paper should be stated more clearly and explicitly. The innovation points and contribution statements at the end of the introduction section are suggested to be further refined and condensed to highlight the core values of this paper.

Author response:

Thank you for your review and valuable suggestions. The contribution statement in the original manuscript was somewhat lengthy and did not highlight the core value of this work. In the revised version, we have thoroughly restructured and refined the contribution section at the end of the introduction. The specific modifications are as follows:

(1)We propose a novel knowledge distillation training framework that divides the training process into two stages, enabling the student model to effectively learn comprehensive static knowledge and diverse, dynamic knowledge rather than relying exclusively on the teacher model.

(2)We introduce heterogeneous branches and a Progressive Feature Fusion (PFF) module, which endows the branches with distinct abilities while efficiently fusion differentiated feature representations, thereby enhancing the diversity of dynamic knowledge.

(3)Extensive comparative experiments demonstrate the superior performance of the proposed THFKD across various datasets and network architectures. For instance, on Tiny-ImageNet, THFKD achieves a 0.71% improvement over state-of-the-art (SOTA) methods.

3.More high-quality papers need to be added to further highlight the value of the proposed method, such as the introduction or related work chapters, including but not limited to the following papers:

[1] Knowledge distillation: A survey[J]. International Journal of Computer Vision, 2021, 129(6): 1789-1819.

[2] Learning from Human Educational Wisdom: A Student-Centered Knowledge Distillation Method[J]. IEEE Transactions on Pattern Analysis and Machine Intelligence, 2024, 46(6): 4188-4205.

[3] Improving Knowledge Distillation via Head and Tail Categories[J]. IEEE Transactions on Circuits and Systems for Video Technology, 2024, 34(5): 3465-3480.

and so on…

Author response:

Thank you for your review and valuable suggestions. Following your recommendations, we have added relevant citations in the introduction and related work sections and further optimized the organization of references to more clearly highlight the value of our proposed method.

[1] Knowledge distillation: A survey[J]. International Journal of Computer Vision, 2021, 129(6): 1789-1819.

[2]Xin X, Song H, Gou J. A new similarity-based relational knowledge distillation method[C]//ICASSP 2024-2024 IEEE International Conference on Acoustics, Speech and Signal Processing (ICASSP). IEEE, 2024: 3535-3539.

4.https://github.com/GitLpf/THFKD. The github URL provided by the author is missing the necessary code and other information. However, one of the most valuable contributions of this paper is the availability of their open-source code. Therefore, it is strongly recommended that the authors carefully organize the relevant de-sensitization data, code, and processing flow introduction into github or other open-source forms, and give them in the form of URL links in the Abstract, so as to enhance the influence of the text work and attract more interested readers to follow the work of this paper.

Author response:

Thank you for your recognition and valuable suggestions. We completely agree that the open-source code is one of the key contributions of this work. After receiving your feedback, we have updated the GitHub repository to include the necessary code and data.

5.In the Section of Methodology, the two-stage optimization strategy and progressive feature fusion module should be described in more detail with mathematical formulations.

Author response:

Thank you for your valuable suggestions regarding the methodology section. Based on your feedback, we have provided a detailed mathematical formulation of the two-stage optimization strategy and the progressive feature fusion module in the revised version. The specific additions are as follows:

(1)The feature map of the first branch is used as the initial input fused feature, i.e., . The expression for the features outputted by subsequent progressive fusion is:

7

8

where is the fusion function from the Progressive feature fusion module, which is iteratively utilized throughout the training process to progressively amalgamate features. For instance, with the number of branches in the model set to four, the ultimate fused feature is denoted as , where the fusion process is articulated as .

(2) �13

where determines the duration of the weight change period, and is the -th epoch with the rate of weight increase being smoothed through the empirical application of a factor of -5. During the training process, the initial parameter is utilized to regulate the epoch at which the ramp-up weight commences, with the setting range of being between 0 and 300.

6.In the Section of Methodology, the loss functions and training procedure should be explained more thoroughly.

Author response:

Thank you for your valuable suggestions regarding the methodology section. Based on your feedback, we have modified and added to the loss formula and the training process in the revised version.

7.In the Section of Experiments, The experimental setup (datasets, model architectures, hyper parameters) should be described in more detail.

Author response:

Thank you for your review and valuable suggestions. We have followed the standard experimental setup in the field of knowledge distillation, including the selection of datasets, model architectures, and hyperparameters. To ensure fairness, we have used hyperparameter settings consistent with existing knowledge distillation research, such as temperature coefficient, optimizer, and learning rate scheduling strategies. We have also introduced the relevant literature in the experimental setup section.

8.More ablation studies should be added to analyze the impact of different components (e.g. fusion module, loss weights). Moreover, more tables are suggested to be added to visually present the comparison results of the performance before and after ablation.

Author response:

Thank you for your review and valuable suggestions. In the revised version, we have added ablation experiments on the depth and width of the heterogeneous branches to analyze their impact further. The details are as follows:

To analyze the impact of heterogeneous branches (in terms of width and depth), we conducted experiments on the ResNet32 model by varying the number of auxiliary blocks and the channel size. Specifically, the number of auxiliary blocks was set to 0–2(NA = {0,1,2}), where NA = 0 indicates that each branch has the same structure without auxiliary blocks, and NA = 1 corresponds to the original experimental setting of THFKD. The channel size of the auxiliary blocks was set to either 1× or 2× the number of channels in the high-level layers (CA = {}), indicating that each branch has the same capacity. As shown in Table 8, the experimental results demonstrate that the number of auxiliary blocks (NA) and the channel size (CA) impact model performance differently.

First, regarding the number of auxiliary blocks (NA), the homogeneous branches without auxiliary blocks (NA = 0) exhibited the lowest accuracy (74.3%–74.6%), indicating that the model’s feature representation and diversity were constrained. When NA =2, the accuracy of each branch improved (74.8%–75.0%); however, the performance of the more complex Branch-4 was inferior to that of Branch-3. This suggests that continuously increasing the number of auxiliary blocks does not necessarily enhance performance, and an overly complex structure may hinder model optimization. In contrast, the THFKD setting (NA = 1) achieved the best performance (75.0%–76.6%), with branch performance progressively improving, thereby validating the rationality of the heterogeneous branch design.

Regarding the impact of channel size (CA), when the channels were doubled (CA = ), the accuracy of Branch-2 and Branch-3 increased to 75.77% and 75.90%, respectively. However, the accuracy of Branch-1 is similar under both CA = and CA = , indicating that uniform channel expansion (i.e., identical auxiliary blocks) led to branches with similar abilities, failing to capture diverse feature representations. Consequently, the dynamic knowledge lacked sufficient diversity. The performance of THFKD (75.03%–76.61%) further confirms the effectiveness of the heterogeneous branch design.

Table.8 Accuracy(%) of branches with different abilities on CIFAR-100.

B Branch-1 Branch-2 Branch-3 Branch-4

NA=0 74.36 74.49 74.33 74.65

NA=2 74.83 75.02 75.64 75.06

CA= 74.51 74.71 75.16 75.15

CA= 74.44 75.77 75.90 75.74

THFKD(NA=1) 75.03 75.36 75.20 76.61

Additionally, we have reported ablation experiments on the fusion module and loss weights in the manuscript, which correspond to the loss term in Figure 4 and the experimental tables for parameters and .

9.The results should be analyzed in more depth, with insights into why the proposed method works well.

Author response:

Thank you for your review and valuable suggestions. We noticed that the analysis of the experimental results in the original manuscript was not sufficiently in-depth. Therefore, in the revised version, we have provided a clearer explanation of the validity of the experimental results. For example, regarding the analysis of Table 5:

We compared the online knowledge distillation methods based on multi-branch models, including CL, ONE, and OKDDip(all configured with four branches during training), with AFID, which employs dual independent models. As shown in Table 5, the first row presents the baseline classification accuracies of the high-parameter models ResNet8x4 and WRN16-2, which are 72.50% and 73.26%, respectively, both higher than the 71.14% accuracy of ResNet32. However, when trained using CL, ONE, and OKDDip, the classification accuracies of these high-parameter models (ResNet8x4 and WRN16-2) decrease compared to ResNet32 (72.56% > 61.81% and 72.43%; 73.50% > 61.66% and 72.86%; 74.32% > 72.75% and 73.96%). Notably, in the AFID, the independent model training approach does not exhibit such accuracy degradation (74.05% < 74.56% and 74.94%). In contrast, THFKD shows a 1.09% improvement over OKDDip when using ResNet8x4 and a 1.37% improvement over AFID when using WRN16-2 while avoiding the phenomenon where high-parameter models (ResNet8x4 and WRN16-2) underperform ResNet32 in classification accuracy. This demonstrates that THFKD adapts to various model architectures and effectively mitigates the problem of homogenization.

Table4.Compare the results of the OKD method on CIFAR-100.

Student ResNet32

71.14 ResNet8x4

72.50 WRN16-2

73.26

CL 72.56 61.81 72.43

ONE 73.50 61.66 72.86

OKDDip 74.32 72.75 73.96

AFID 74.05 74.56 74.94

THFKD 75.41 75.82 76.31

10.In the Section of Conclusion, the conclusion should summarize the key findings and contributions. Limitations of the current work and potential future directions should be discussed.

Author response:

Thank you for your review and valuable suggestions. In the revised version, we have improved the conclusion section to summarize the key findings and contributions more clearly. Additionally, we have added an analysis of the limitations of the current method and discussed potential future research directions to provide a more comprehensive discussion. The specific modifications are as follows:

We propose a novel Two-stage optimization based on heterogeneous branch fusion for knowledge distillation (THFKD). In traditional knowledge distillation, the student model is solely guided by the fixed knowledge of the teacher model, which cannot supplement or expand upon this knowledge, thereby limiting the student's generalization capability. In contrast, our student model leverages heterogeneous branches and a Progressive Feature Fusion (PFF) module to generate collaborative learning objectives, enabling the transfer of diversified, dynamic knowledge rather than relying exclusively on the teacher model. During training, the dynamic knowledge is combined with the static knowledge from the pre-trained teacher, while the two-stage optimization strategy provides appropriate knowledge at different training stages, enhancing the student model's learning efficiency and generalization ability. Extensive experiments were conducted on both standard datasets and long-tailed datasets, demonstrating that THFKD achieves superior performance compared to state-of-the-art methods. This demonstrates its broad application potential in image classification tasks. Additionally, we provide an analysis and visualization to further understand the underlying mechanisms of the proposed method. However, our method also exhibits certain limitations. THFKD performs slightly below existing approaches on lightweight models such as ShuffleNet, and the range of model combinations utilized is not yet comprehensive. In future work, we aim to investigate these limitations further and extend THFKD to other domains, such as object detection and image segmentation, to explore its broader applicability.

11.The writing should be polished for clarity, grammar, and flow. Some sentences are overly long and could be broken up. Consistent terminology should be used throughout (e.g. "student model" vs "target model").

Author response:

Thank you for your review and valuable suggestions. In the revised version, we have optimized the clarity of the writing to make the expression smoother. Additionally, we have standardized the terminology, such as using "student model" throughout the manuscript instead of "target model," to ensure consistency in the terminology.

Reviewer #2:

1.The authors present nine figures in this paper. Unfortunately, the way these figures are presented significantly hinders the readability of the paper. The captions, without the corresponding figures, are scattered throughout the paper, preventing readers from reading the captions while simultaneously viewing the corresponding figures. All nine figures related to these captions are located at the very end of the paper. Moreover, the figure corresponding to the caption ‘Fig 1. Two training frameworks. (a) using the OKD method and (b) using the KD method.’ is placed as the eighth figure in the list of nine figures located at the end of the paper. Thus, the order of the captions throughout the paper is completely different from the order of the figures at the end. Additionally, the eighth figure in this list contains two parts: ‘(a) Multi-branch OKD’ and ‘(a) Multi-branch KD’, with each part labeled as (a), which is inconsistent with the caption for Figure 1. This should be corrected. Furthermore, each figure should be positioned alongside its caption so that readers can read the figure descriptions while simultaneously viewing the corresponding figures. This will enable readers to easily correlate, for example, the notation used in the captions with

---

## [Decision Letter · Decision Letter 1]

15 Apr 2025

PONE-D-25-00345R1Two-stage optimization based on heterogeneous branch fusion for knowledge distillationPLOS ONE

Dear Dr. Zhang,

Thank you for submitting your manuscript to PLOS ONE. After careful consideration, we feel that it has merit but does not fully meet PLOS ONE’s publication criteria as it currently stands. Therefore, we invite you to submit a revised version of the manuscript that addresses the points raised during the review process.

The reviewers raised minor comments that need to be addressed.

We look forward to receiving your revised manuscript.

Kind regards,

Alberto Marchisio

Academic Editor

PLOS ONE

Journal Requirements:

Reviewers' comments:

Reviewer's Responses to Questions

**Comments to the Author**

1. If the authors have adequately addressed your comments raised in a previous round of review and you feel that this manuscript is now acceptable for publication, you may indicate that here to bypass the “Comments to the Author” section, enter your conflict of interest statement in the “Confidential to Editor” section, and submit your "Accept" recommendation.

Reviewer #1: (No Response)

Reviewer #2: All comments have been addressed

Reviewer #3: (No Response)

Reviewer #4: All comments have been addressed

Reviewer #5: (No Response)

Reviewer #6: All comments have been addressed

2. Is the manuscript technically sound, and do the data support the conclusions?

Reviewer #1: Yes

Reviewer #2: Yes

Reviewer #3: Yes

Reviewer #4: Yes

Reviewer #5: Yes

Reviewer #6: Yes

3. Has the statistical analysis been performed appropriately and rigorously? 

Reviewer #1: Yes

Reviewer #2: Yes

Reviewer #3: Yes

Reviewer #4: No

Reviewer #5: N/A

Reviewer #6: Yes

4. Have the authors made all data underlying the findings in their manuscript fully available?

Reviewer #1: Yes

Reviewer #2: Yes

Reviewer #3: Yes

Reviewer #4: Yes

Reviewer #5: Yes

Reviewer #6: Yes

5. Is the manuscript presented in an intelligible fashion and written in standard English?

Reviewer #1: Yes

Reviewer #2: Yes

Reviewer #3: Yes

Reviewer #4: Yes

Reviewer #5: Yes

Reviewer #6: Yes

6. Review Comments to the Author

Reviewer #1: Thanks to the authors for their careful revision, some issues need to be corrected further to improve the quality and readability of this article.

Innovativeness and Contributions

1.Elaborate on the Unique Advantages of the Innovations (Paragraph: Lines 66-69)

The authors propose heterogeneous branch structures and a progressive feature fusion module but need to provide a more detailed comparison with existing technologies. For instance, in lines 66-69, the authors mention, “We propose a novel knowledge distillation training framework,” yet the specific differences between this framework and existing two-stage training methods (such as conventional KD and OKD) are not adequately highlighted.

Experimental Design and Results Analysis

2.Supplement Comparative Experiments with the Latest Research (Section: Experimental Results)

In the experimental results section (e.g., Tables 1 and 2), the authors primarily compare their method with classical approaches such as KD, FitNet, and CRD. It is suggested to supplement the experiments with comparisons to the latest knowledge distillation methods from 2023 to 2025, particularly on large-scale datasets like Tiny-ImageNet. For example, the authors could reference the following high-quality literature published in IEEE Transactions:

"A Lightweight Residual Network Based on Improved Knowledge Transfer and Quantized Distillation for Cross-Domain Fault Diagnosis of Rolling Bearings" (Expert Systems with Applications, 2024)

"Reciprocal Teacher-Student Learning via Forward and Feedback Knowledge Distillation" (IEEE Transactions on Multimedia, 2024)

"Hvdistill: Transferring Knowledge from Images to Point Clouds via Unsupervised Hybrid-View Distillation" (International Journal of Computer Vision, 2024)

3.In-Depth Analysis of the Intrinsic Reasons for Experimental Results (Section: Table 3 and Related Analysis)

In Table 3, THFKD achieves an accuracy of 58.1% on the CIFAR100-LT dataset, but the reasons for this superior performance on long-tailed datasets are not thoroughly explained. It is recommended to add an analysis discussing how heterogeneous branches and progressive feature fusion assist the model in better learning the features of tail classes under long-tailed distributions. For instance, the authors could combine Grad-CAM visualization results to demonstrate how the model focuses on the key feature regions of tail classes.

4.Add an Analysis of the Trade-off Between Model Complexity and Performance (Section: Experimental Settings)

In the experimental settings section (lines 313-324), the authors mention the use of different model architectures but do not delve into the trade-off between model complexity and performance. It is suggested to add experiments showcasing the performance comparison between THFKD and other methods under different computational resource constraints (such as varying GPU memory usages), as well as how to select the appropriate number of branches and auxiliary block configurations. For example, a table could be added to display the accuracy and memory usage of different branch numbers (2, 3, 4, 5) on the CIFAR-100 and Tiny-ImageNet datasets.

Writing Quality and Expression

5.Optimize the Article Structure and Logical Flow (Section: Related Work)

In the related work section (lines 73-140), the authors list various knowledge distillation methods but lack a clear classification and organization. It is suggested to categorize the introduction according to “methods based on static knowledge,” “methods based on dynamic knowledge,” and “hybrid methods” to facilitate readers’ understanding of the characteristics of different methods and THFKD’s positioning. For example:

Methods based on static knowledge: including conventional KD, FitNet, etc.

Methods based on dynamic knowledge: including DML, ONE, OKDDip, etc.

Hybrid methods: methods that combine static and dynamic knowledge, such as THFKD.

6.Refine Language Expression (Throughout the Article)

It is recommended that the authors refine the language throughout the article to ensure concise and clear expression. Additionally, it is suggested that the authors engage English-native experts or professional language editing services to further enhance the quality of the language.

7.Code and Documentation Optimization

Enrich and Optimize Code and Documentation (Section: GitHub Link)

The GitHub link has been adequately supplemented, but it is recommended to further enrich and optimize the code and documentation to assist readers in better and faster reproduction and citation of the work. For example:

Provide detailed instructions in the README document on how to install dependencies, run the code, and reproduce experimental results.

Offer example code and configuration files demonstrating how to use the THFKD method for training and testing.

Add code comments explaining the functions of key modules and functions to help readers better understand the code implementation.

Reviewer #2: The authors investigate knowledge distillation methods and propose a two-stage optimization based on heterogeneous branch fusion for knowledge distillation. The paper is interesting and addresses intriguing questions. The authors conduct experiments rigorously and draw conclusions based on the presented data. Their analysis is thorough and the paper is presented in an intelligible manner. All my concerns have been addressed in the revised version.

Reviewer #3: 1. Author's response to this reviewers comments

The author's have adequately responded to this reviewer's comments, thank you. The availability of the code on Git allows interested readers to clarify architectural questions that may remain after reading the paper.

2. Technical question

2.1 Clarity of Progressive Feature Fusion (PFF) Module: In the description of the PFF module, please elaborate on the exact role/calculation of the scalar parameter 'g' mentioned in Eq. 8. How is 'g' determined or learned?

Reviewer #4: The manuscript has been significantly improved based on the reviewers' feedback. The authors have been responsive and provided detailed justifications and revisions. The remaining concern regarding statistical analysis was addressed by removing the claims, which is an acceptable, though less ideal, resolution.

Reviewer #5: SUMMARY: Most of my concerns have been addressed by the authors. However, I think the spirit of the conclusions about the benefits of their proposed THFKD method is too optimistic. In essence, it is not really clear that the THFKD method is superior to the SOTA methods because the performance improvement is marginal (less than 1%, as indicated in the abstract when the 0.71% performance increase is mentioned). Without a statistical analysis of the significance of the performance differences observed, the improvement achieved by the THFKD method is not conclusive, and it can even be due to noise.

In what follows, I suggest specific modifications in the manuscript with the objective of mitigating the excessive optimism expressed with regard to the benefits of the THFKD method over the SOTA methods.

DETAILED COMMENTS:

A few suggestions for improvement are in order of their appearances in the manuscript:

1) In the abstract we read "THFKD effectively improves...". I recommend changing the whole sentence starting at "Experiments results ... generalization ability" to "Although no tests of statistical significance were carried out, our experimental results on standard datasets (CIFAR-100, Tiny-ImageNet) and long-tail datasets (CIFAR100-LT) suggest that THFKD may slightly improve the student model’s classification accuracy and generalization ability."

The reason for this change is that the increase in accuracy is really marginal, as e.g. stated in the phrase following the above sentence, where the improvement in accuracy is quantified as just 0.71%. The improvement is so small that we cannot be certain that it is an actual signal, or it is something obtained just by chance, as no statistical analysis was carried out to confirm it.

2) Line 176: The description of the "y" labels was improved to indicate they correspond to one-hot encoding of the actual label used to classify the images. However, I still suggest adding a short phrase to the sentence to make the definition of one-hot encoding complete, namely by adding the phrase after the comma in the following sentence: "with its corresponding one-hot label y = [y1, ..., yK], where y1, ..., yK in {0, 1} and sum_k {yk} = 1".

3) Although no statistical analysis has been performed to quantify the significance of the differences in accuracy observed across methods, almost all mentioning to "a significant difference" were correctly removed from the text.

However, there is still one use of "significant" which I suggest replacing with a quantitative evaluation: in line 432 we read "THFKD exhibits significant performance improvement when the number of branches is set to 4." I suggest replacing this sentence with "THFKD exhibits a performance improvement of about 2% (from ~ 73% to 75%) when the number of branches is increased from 1 to 4.". Again, no statistical test was carried out to evaluate whether this increase in performance is statistically significant, this is the reason for removing the word "significant" and providing instead the actual quantitative change observed in the specific experiment carried out by the authors.

4) Lines 448-475: I find the new quantitative analysis on the impact of heterogeneous branches a little "over-conclusive". As seen in Table 9, the differences in accuracy when varying parameters NA and CA is always VERY small: except for 5 values of the 20 values reported (namely 75.64, 75.77, 75.90, 75.74, 76.61), the difference between any of the accuracy values and the smallest accuracy for the "NA=0, Branch-1" case (= 74.36) is at most 1%. That is, the effect of branch heterogeneity doesn't seem to be very strong... The observed differences could just be noise. I don't find the conclusion in the last line ("The performance of THFKD (75.03%–76.61%) further confirms the effectiveness of the heterogeneous branch design.") really correct. The accuracy even decreases when going from branch-2 to branch-3 from 75.36 to 75.20, suggesting that no strong signal about accuracy improvement is present in the reported values...

I suggest thoroughly re-writing all three paragraphs in view of the above, and avoid conveying that there is a strong effect on accuracy by branch heterogeneity.

5) Lines 450-451: The acronym "NA" is mentioned but not defined. The sentence should read: "Specifically, the number of auxiliary blocks (NA) was set to ...". The other two repetitions of the definition of NA (in lines 456 and 458) should be removed because NA is now defined on line 450.

6) Same comment for the "CA" acronym applies as for the "NA" acronym: CA should be defined the first time "channel size" is mentioned, namely on line 453, and then be removed from their redefinition below (lines 456 and 468).

7) Conclusion: Although the authors removed their initially (apparently arbitrary) selection of the quantitative results to report from the Conclusion section, the new sentence is too vague, namely: "THFKD achieves superior performance compared to state-of-the-art methods". How much superior? I think the superiority is not strong enough and the level of noise in the accuracy results has not been evaluated. This lack of statistical analysis does not allow to write a conclusion that states that THFKD is superior to the SOTA methods. Please see my suggestions for modification of the abstract in item (1) above for a guideline on how to change this conclusion.

Reviewer #6: • The authors have worked hard and incorporated the communicated comments. Hence, the overall quality and readability of the manuscript is enhanced.

• In my opinion, the manuscript may be accepted in the current form.

7. PLOS authors have the option to publish the peer review history of their article (what does this mean?). If published, this will include your full peer review and any attached files.

Reviewer #1: No

Reviewer #2: No

Reviewer #3: No

Reviewer #4: No

Reviewer #5: **Yes: **Daniel Mastropietro

Reviewer #6: No

---

## [Author Response · Author response to Decision Letter 2]

7 May 2025

Dear Editors and Reviewers:

Thank you for your letter and for the reviewers’ comments concerning our manuscript. Those comments are valuable and very helpful for revising and improving our paper, as well as the guiding significance to our studies. We have carefully considered the comments and have made revision which is hoped to meet with approval. In order to highlight the changes that we have made, we additionally submit a marked version of the paper and revised portion which is marked in blue.

Reviewer #1:

1.Elaborate on the Unique Advantages of the Innovations (Paragraph: Lines 66-69)

The authors propose heterogeneous branch structures and a progressive feature fusion module but need to provide a more detailed comparison with existing technologies. For instance, in lines 66-69, the authors mention, “We propose a novel knowledge distillation training framework,” yet the specific differences between this framework and existing two-stage training methods (such as conventional KD and OKD) are not adequately highlighted.

Author response:

Thank you for your review and valuable suggestions. we have refined the contribution section at the end of the introduction. The specific modifications are as follows:

1.We propose a novel knowledge distillation training framework. Unlike conventional approaches, our framework enables the student model to learn not only from a pre-trained teacher model but also to acquire dynamic knowledge from heterogeneous branches. The training process consists of two stages: first establishing foundational cognition through comprehensive static knowledge, then introducing diversified dynamic knowledge to achieve synergistic optimization between static and dynamic knowledge.

2.Supplement Comparative Experiments with the Latest Research (Section: Experimental Results)

In the experimental results section (e.g., Tables 1 and 2), the authors primarily compare their method with classical approaches such as KD, FitNet, and CRD. It is suggested to supplement the experiments with comparisons to the latest knowledge distillation methods from 2023 to 2025, particularly on large-scale datasets like Tiny-ImageNet. For example, the authors could reference the following high-quality literature published in IEEE Transactions:

"A Lightweight Residual Network Based on Improved Knowledge Transfer and Quantized Distillation for Cross-Domain Fault Diagnosis of Rolling Bearings" (Expert Systems with Applications, 2024)

"Reciprocal Teacher-Student Learning via Forward and Feedback Knowledge Distillation" (IEEE Transactions on Multimedia, 2024)

"Hvdistill: Transferring Knowledge from Images to Point Clouds via Unsupervised Hybrid-View Distillation" (International Journal of Computer Vision, 2024)

Author response:

Thank you for this valuable suggestion. We agree that incorporating comparisons with more recent knowledge distillation methods is important to demonstrate the competitiveness of our approach. In the revised manuscript, we have updated Tables 2 to include results from state-of-the-art distillation methods FFKD published 2024, on the Tiny-ImageNet dataset

Table2. Accuracy(%) of KD Methods on Tiny-ImageNet. ResNet34-ResNet18.

Teacher Student KD[11] AT[26] CRD[23] AFD[41] FFKD[42] NRKD[30] THFKD

66.84 65.14 67.66 67.76 67.94 68.10 68.51 68.93 69.64

[42] Reciprocal Teacher-Student Learning via Forward and Feedback Knowledge Distillation" (IEEE Transactions on Multimedia, 2024)

3.In-Depth Analysis of the Intrinsic Reasons for Experimental Results (Section: Table 3 and Related Analysis)

In Table 3, THFKD achieves an accuracy of 58.1% on the CIFAR100-LT dataset, but the reasons for this superior performance on long-tailed datasets are not thoroughly explained. It is recommended to add an analysis discussing how heterogeneous branches and progressive feature fusion assist the model in better learning the features of tail classes under long-tailed distributions. For instance, the authors could combine Grad-CAM visualization results to demonstrate how the model focuses on the key feature regions of tail classes.

Author response:

Thank you for your review and valuable suggestions. We have supplemented the explanation for the superior performance of THFKD on long-tailed datasets in Table 3. The specific modifications are as follows:

A long-tailed dataset has a highly imbalanced sample distribution, where the number of samples in a few categories is significantly larger than that in most categories. As shown in Table3, although THFKD is not explicitly designed for long-tail classification, it has demonstrated strong classification performance on the long-tailed dataset CIFAR100-LT. The heterogeneous branches designed in this paper allow each branch to specialize in different categories and data characteristics, thereby enhancing the model’s ability to capture information from tail classes. Building on this architecture, our progressive feature fusion module enables effective integration of differentiated features, while the knowledge distillation mechanism facilitates knowledge sharing and complementarity specifically for tail categories. This indicates that the student model is capable of learning comprehensively across all categories, avoiding bias toward the more frequent classes in the early stages of training, and further optimizing the model with diversified knowledge to enhance its generalization ability, thereby improving the classification performance of the model on the tail categories.

4.Add an Analysis of the Trade-off Between Model Complexity and Performance (Section: Experimental Settings)

In the experimental settings section (lines 313-324), the authors mention the use of different model architectures but do not delve into the trade-off between model complexity and performance. It is suggested to add experiments showcasing the performance comparison between THFKD and other methods under different computational resource constraints (such as varying GPU memory usages), as well as how to select the appropriate number of branches and auxiliary block configurations. For example, a table could be added to display the accuracy and memory usage of different branch numbers (2, 3, 4, 5) on the CIFAR-100 and Tiny-ImageNet datasets.

Author response:

We sincerely appreciate the reviewers' valuable suggestions regarding the model complexity analysis. Due to limited available equipment and GPU resources, we are currently unable to conduct experiments under different computational constraints. We fully agree that performance comparisons under varying resource limitations would be valuable, and we will make every effort to acquire additional hardware in the future to conduct these experiments.We have provided ablation studies on both the number of branches and auxiliary blocks in Figure 7 and Table 9, respectively.

5.Optimize the Article Structure and Logical Flow (Section: Related Work)

In the related work section (lines 73-140), the authors list various knowledge distillation methods but lack a clear classification and organization. It is suggested to categorize the introduction according to “methods based on static knowledge,” “methods based on dynamic knowledge,” and “hybrid methods” to facilitate readers’ understanding of the characteristics of different methods and THFKD’s positioning. For example:

Methods based on static knowledge: including conventional KD, FitNet, etc.

Methods based on dynamic knowledge: including DML, ONE, OKDDip, etc.

Hybrid methods: methods that combine static and dynamic knowledge, such as THFKD.

Author response:

Thank you for your valuable suggestions regarding the methodology section. As highlighted by the reviewer, a well-structured taxonomy can better elucidate the methodological evolution and our contribution.

Adopting the three-level classification system as suggested, existing methods have been reorganized into:

Methods based on static knowledge (Traditional KD, FitNet, etc.)

Methods based on dynamic knowledge (DML, ONE, OKDDip, etc.)

Multi-branch architectures and feature fusion

Terminology Retention Statement:

We retain "Multi-branch architectures and feature fusion" as the third category title because this terminology more precisely reflects the core innovation of our method.

6.Refine Language Expression (Throughout the Article)

It is recommended that the authors refine the language throughout the article to ensure concise and clear expression. Additionally, it is suggested that the authors engage English-native experts or professional language editing services to further enhance the quality of the language.

Author response:

Thank you for your thorough review of our paper and valuable suggestions. We have followed your advice and reviewed and improved the grammar, spelling, and sentence structure throughout the manuscript.

7.Code and Documentation Optimization

Enrich and Optimize Code and Documentation (Section: GitHub Link)

The GitHub link has been adequately supplemented, but it is recommended to further enrich and optimize the code and documentation to assist readers in better and faster reproduction and citation of the work. For example:

Provide detailed instructions in the README document on how to install dependencies, run the code, and reproduce experimental results.

Offer example code and configuration files demonstrating how to use the THFKD method for training and testing.

Add code comments explaining the functions of key modules and functions to help readers better understand the code implementation.

Author response:

Thank you for your thorough review of our paper and valuable suggestions. We have provided detailed code execution and experimental result reproduction instructions in the README documentation, along with code comments explaining key modules and loss functions, to facilitate readers' understanding of the implementation.

Reviewer #3:

1.Clarity of Progressive Feature Fusion (PFF) Module: In the description of the PFF module, please elaborate on the exact role/calculation of the scalar parameter 'g' mentioned in Eq. 8. How is 'g' determined or learned?

Author response:

Thank you for your review and valuable suggestions. A detailed explanation of the scalar parameter g is as follows:

In the fusion process described in Equation (8), as a balancing factor, dynamically weighting the contributions of the current branch feature and the fused features from the previous layer, with parameter is a scalar parameter with a value range of 0 to 1, when approaches 1, the current branch features dominates the fusion output, when approaches 0, the fused features from the previous level receive higher weight. In our method, is set to 0.5, ensuring equal importance of both current branch features and prior fused features. We have provided a more comprehensive explanation of this mechanism in the revised manuscript.

Reviewer #5:

1.In the abstract we read "THFKD effectively improves...". I recommend changing the whole sentence starting at "Experiments results ... generalization ability" to "Although no tests of statistical significance were carried out, our experimental results on standard datasets (CIFAR-100, Tiny-ImageNet) and long-tail datasets (CIFAR100-LT) suggest that THFKD may slightly improve the student model’s classification accuracy and generalization ability."

The reason for this change is that the increase in accuracy is really marginal, as e.g. stated in the phrase following the above sentence, where the improvement in accuracy is quantified as just 0.71%. The improvement is so small that we cannot be certain that it is an actual signal, or it is something obtained just by chance, as no statistical analysis was carried out to confirm it.

Author response:

Thank you for your important reminder regarding the rigor of experimental results presentation. We have fully adopted your suggestions and revised the abstract as follows:

Although no tests of statistical significance were carried out, our experimental results on standard datasets (CIFAR-100, Tiny-ImageNet) and long-tail datasets (CIFAR100-LT) suggest that THFKD may slightly improve the student model’s classification accuracy and generalization ability. For instance, using ResNet18 on the Tiny-ImageNet dataset, the accuracy reaches 69.64%, a 0.71% improvement over the state-of-the-art (SOTA).

2.Line 176: The description of the "y" labels was improved to indicate they correspond to one-hot encoding of the actual label used to classify the images. However, I still suggest adding a short phrase to the sentence to make the definition of one-hot encoding complete, namely by adding the phrase after the comma in the following sentence: "with its corresponding one-hot label y = [y1, ..., yK], where y1, ..., yK in {0, 1} and sum_k {yk} = 1".

Author response:

Thank you for your rigorous requirements regarding the mathematical notation definitions. We have made the following improvements to Line 176:

Given a dataset with classes, where , and a training sample with its corresponding one-hot label , where and , when is input into the student model, the output probability of the -th branch for class is calculated as:

3.Although no statistical analysis has been performed to quantify the significance of the differences in accuracy observed across methods, almost all mentioning to "a significant difference" were correctly removed from the text.

However, there is still one use of "significant" which I suggest replacing with a quantitative evaluation: in line 432 we read "THFKD exhibits significant performance improvement when the number of branches is set to 4." I suggest replacing this sentence with "THFKD exhibits a performance improvement of about 2% (from ~ 73% to 75%) when the number of branches is increased from 1 to 4.". Again, no statistical test was carried out to evaluate whether this increase in performance is statistically significant, this is the reason for removing the word "significant" and providing instead the actual quantitative change observed in the specific experiment carried out by the authors.

Author response:

Thank you for your persistent suggestions regarding terminology refinement. We have implemented the following key improvements to Line 432:

We set the total number of branches to range from 2 to 6 to analyze the impact of branch quantity on performance. As shown in Fig.7, THFKD exhibits a performance improvement of about 2% (from 73% to 75%) when the number of branches is increased from 1 to 4.

4.(1)Lines 448-475: I find the new quantitative analysis on the impact of heterogeneous branches a little "over-conclusive". As seen in Table 9, the differences in accuracy when varying parameters NA and CA is always VERY small: except for 5 values of the 20 values reported (namely 75.64, 75.77, 75.90, 75.74, 76.61), the difference between any of the accuracy values and the smallest accuracy for the "NA=0, Branch-1" case (= 74.36) is at most 1%. That is, the effect of branch heterogeneity doesn't seem to be very strong... The observed differences could just be noise. I don't find the conclusion in the last line ("The performance of THFKD (75.03%–76.61%) further confirms the effectiveness of the heterogeneous branch design.") really correct. The accuracy even decreases when going from branch-2 to branch-3 from 75.36 to 75.20, suggesting that no strong signal about accuracy improvement is present in the reported values...

I suggest thoroughly re-writing all three paragraphs in view of the above, and avoid conveying that there is a strong effect on accuracy by branch heterogeneity.

(2) Lines 450-451: The acronym "NA" is mentioned but not defined. The sentence should read: "Specifically, the number of auxiliary blocks (NA) was set to ...". The other two repetitions of the definition of NA (in lines 456 and 458) should be removed because NA is now defined on line 450.

(3) Same comment for the "CA" acronym applies as for the "NA" acronym: CA should be defined the first time "channel size" is mentioned, namely on line 453, and then be removed from their redefinition below (lines 456 and 468).

Author response:

We sincerely appreciate your meticulous review of our experimental analysis and terminology standardization. We have rigorously implemented your sugges

---

## [Decision Letter · Decision Letter 2]

4 Jun 2025

Two-stage optimization based on heterogeneous branch fusion for knowledge distillation

PONE-D-25-00345R2

Dear Dr. Zhang,

We’re pleased to inform you that your manuscript has been judged scientifically suitable for publication and will be formally accepted for publication once it meets all outstanding technical requirements.

Kind regards,

Alberto Marchisio

Academic Editor

PLOS ONE

Additional Editor Comments (optional):

Reviewers' comments:

Reviewer's Responses to Questions

**Comments to the Author**

1. If the authors have adequately addressed your comments raised in a previous round of review and you feel that this manuscript is now acceptable for publication, you may indicate that here to bypass the “Comments to the Author” section, enter your conflict of interest statement in the “Confidential to Editor” section, and submit your "Accept" recommendation.

Reviewer #1: All comments have been addressed

Reviewer #3: All comments have been addressed

Reviewer #5: All comments have been addressed

2. Is the manuscript technically sound, and do the data support the conclusions?

Reviewer #1: Yes

Reviewer #3: Yes

Reviewer #5: Yes

3. Has the statistical analysis been performed appropriately and rigorously? 

Reviewer #1: Yes

Reviewer #3: No

Reviewer #5: No

4. Have the authors made all data underlying the findings in their manuscript fully available?

Reviewer #1: Yes

Reviewer #3: Yes

Reviewer #5: Yes

5. Is the manuscript presented in an intelligible fashion and written in standard English?

Reviewer #1: Yes

Reviewer #3: Yes

Reviewer #5: Yes

6. Review Comments to the Author

Reviewer #1: The issues raised have been properly addressed. Thus, the paper is recommended for publication in the journal.

Reviewer #3: (No Response)

Reviewer #5: All my comments have been addressed as well as those posed by other reviewers.

Nevertheless, two comments to take into account before the final version for publication:

1) One small change to do:

Please replace the following phrase in the Notation subsection,

"... where y1, . . . , yK ∈ (0, 1) and sum{k} y(k) = 1."

with this one:

"... where y1, . . . , yK ∈ {0, 1} and sum{k} y(k) = 1."

where the interval (0, 1) has been replaced with the finite set {0, 1}.

Recall that one-hot encoding implies that each y(k) takes either the value 0 or 1, and not any real value in between.

2) One clarification to add:

In the "Branches with different abilities" subsection, for the sentence,

"Although the performance improvement is modest, the accuracy variation among Branches 2–4 is relatively large (a maximum difference of 1.41), ...",

please consider stating what the two values in Table 9 involved in that difference are. It took me a while to figure that out (I now see it's 75.20% in Branch-3 column and 76.61% in Branch-4 column).

Thanks for your work.

7. PLOS authors have the option to publish the peer review history of their article (what does this mean?). If published, this will include your full peer review and any attached files.

Reviewer #1: No

Reviewer #3: No

Reviewer #5: **Yes: **Daniel Mastropietro

---

## [Editor Report · Acceptance letter]

PONE-D-25-00345R2

PLOS ONE

Dear Dr. Zhang,

I'm pleased to inform you that your manuscript has been deemed suitable for publication in PLOS ONE. Congratulations! Your manuscript is now being handed over to our production team.

Kind regards,

on behalf of

Dr. Alberto Marchisio

Academic Editor

PLOS ONE